# Observation of charge–parity symmetry breaking in baryon decays

LHCb Collaboration*

The Standard Model of particle physics—the theory of particles and interactions at the smallest scale—predicts that matter and antimatter interact differently due to violation of the combined symmetry of charge conjugation (C) and parity (P). Charge conjugation transforms particles into their antimatter particles, whereas the parity transformation inverts spatial coordinates. This prediction applies to both mesons, which consist of a quark and an antiquark, and baryons, which are composed of three quarks. However, despite having been discovered in various meson decays, CP violation has yet to be observed in baryons, the type of matter that makes up the observable Universe. Here we report a study of the decay of the beauty baryon $\Lambda_b^0$ to the $pK^-\pi^+\pi^-$ final state, which proceeds through $b \to u$ or $b \to s$ quark-level transitions, and its CP-conjugated process, using data collected by the Large Hadron Collider beauty experiment[1] at the European Organization for Nuclear Research (CERN). The results reveal significant asymmetries between the decay rates of the $\Lambda_b^0$ baryon and its CP-conjugated antibaryon, providing, to our knowledge, the first observation of CP violation in baryon decays and demonstrating the different behaviours of baryons and antibaryons. In the Standard Model, CP violation arises from the Cabibbo–Kobayashi–Maskawa mechanism[2], and new forces or particles beyond the Standard Model could provide further contributions. This discovery opens a new path in the search for physics beyond the Standard Model.

In 1928, Dirac proposed a theory of electron motion that predicted the existence of the positron, the antimatter counterpart to the electron[3]. Since then, all antimatter partners of known elementary particles and those of composite particles made of quarks (referred to as hadrons) have been discovered in accelerator-based experiments or cosmic rays[4,5]. Astronomical observations indicate that other stars and planets in the Universe are composed of the same type of matter that constitutes the Solar System, namely protons and neutrons forming nuclei that are orbited by electrons, whereas the amount of antimatter particles is negligible[6].

According to cosmological models, matter and antimatter were created in equal amounts at the Big Bang[6]. Then matter and antimatter mostly annihilated in pairs as the Universe cooled down, with a tiny fraction of matter remaining. The dominance of matter requires the violation of both charge conjugation (C) symmetry and charge conjugation and parity symmetry (CP symmetry) in conjunction with other conditions, as proposed by Sakharov in 1967 (ref. 7). Experimentally, it was established in 1957–1958 that the weak force breaks both parity (P) and C symmetries[8,9]. The violation of the combined CP symmetry was first observed in strange-meson decays in 1964 (ref. 10). This phenomenon was later also observed in beauty-meson decays in 2001 (refs. 11,12) and in charm-meson decays in 2019 (ref. 13). Here, strange, charm and beauty refer to the flavours of the constituent quarks.

Quark dynamics are described by the Standard Model of particle physics. CP violation arises from the Cabibbo–Kobayashi–Maskawa (CKM) mechanism[2]. The CKM mechanism uses a complex 3 × 3 matrix

to describe how quarks of different generations mix under the weak interaction, which is mediated by the exchange of $W^\pm$ bosons. The structure of this mixing is ultimately linked to the Higgs mechanism, which gives rise to the masses of fundamental particles, including the quarks. The matrix contains a non-zero phase parameter, which provides the only known source of CP symmetry breaking. In general, the CKM mechanism is very successful in describing experimental data for CP asymmetries and decay rates[14]. However, the amount of matter–antimatter asymmetry explained by the CKM mechanism is vastly smaller than what astronomical observations indicate, presenting an important challenge to the Standard Model and hinting at the presence of further sources of CP violation[15]. Continuing explorations of CP violation may open new avenues for the discovery of physics beyond the Standard Model.

The lack of observed CP violation in baryons, the predominant form of matter in the visible Universe, remains a puzzle. Similar levels of CP violation in meson and baryon decays are expected due to identical quark-level transitions. Yet, CP violation has so far been detected only in mesons. This discrepancy is especially pronounced in beauty-baryon decays, where large CP asymmetries are anticipated, as seen for beauty mesons. For instance, the beauty-meson decay $B_s^0 \to K^-\pi^+$ shows a $(23.6 \pm 1.7)\%$ CP asymmetry[16,17], whereas the corresponding baryon decays $\Lambda_b^0 \to ph^-$, where $h$ denotes a $K$ or $\pi$ meson, exhibit no such asymmetry with 0.7% precision[18]. Similarly, three-body beauty-meson decays, such as $B^+ \to \pi^+\pi^-\pi^+$, display CP asymmetries of up to 75% (ref. 19), whereas no significant CP violation has been observed

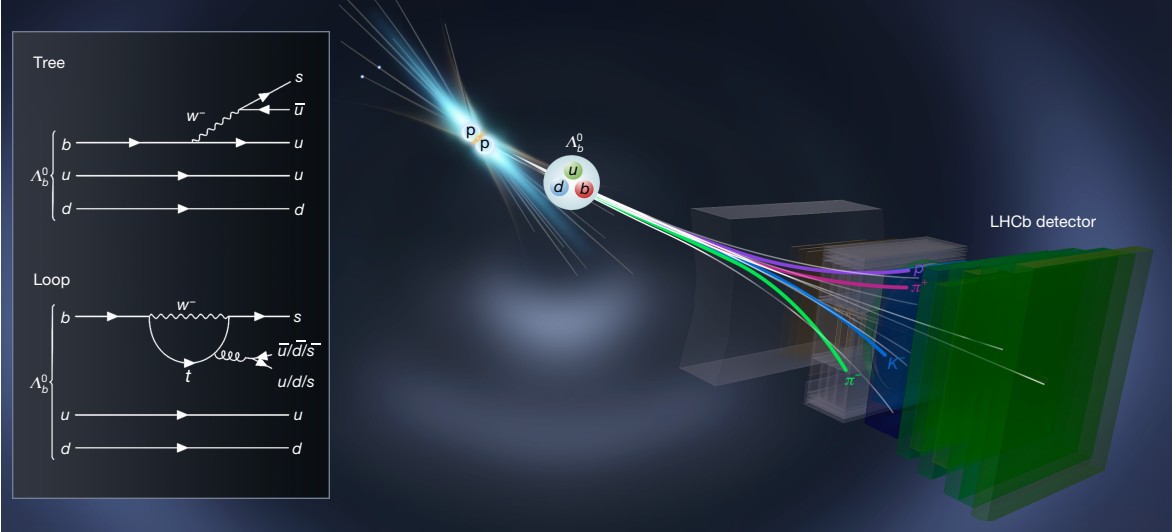

**Fig. 1 | Illustration of $\Lambda_b^0$ production in a $pp$ collision and decay into the $pK^-\pi^+\pi^-$ final state.** The two inset diagrams on the left illustrate the fundamental tree-type and loop-type quark-level processes that mediate the $\Lambda_b^0 \to pK^-\pi^+\pi^-$ decay. The quarks in these processes eventually form $p$, $K^-$, $\pi^+$ and $\pi^-$ particles, combined with further $u\bar{u}$ and $d\bar{d}$ quark pairs created from the vacuum. The final states may also arise through intermediate hadronic resonances. The resulting hadrons were directly detected by the LHCb detector.

in beauty-baryon decays[20,21]. The $\Lambda_b^0 \to \Lambda K^+ K^-$ decay exhibits a hint of CP asymmetries below 20% with a significance of 3.1 standard deviations, requiring further confirmation[22]. No CP violation has been observed in strange and charm baryon decays nor in unflavoured baryon decays.

In this work, we report, to our knowledge, the first observation of CP violation in baryon decays, specifically in the decay of the $\Lambda_b^0$ baryon to a proton, a kaon and a pair of oppositely charged pions, represented as $\Lambda_b^0 \to pK^-\pi^+\pi^-$. This decay proceeds through $b \to u$ or $b \to s$ quark-level transitions, and the measured final-state particles include contributions from various possible intermediate hadronic resonances. In the following, CP-conjugated particles or decays are included if not otherwise specified. The constituents of the $\Lambda_b^0$ baryon are like those of the proton (made of $uud$ quarks), with one of the $u$ quarks replaced by a $b$ quark. The amount of CP violation in $\Lambda_b^0$ decays is quantified by the asymmetry, $\mathcal{A}_{CP}$, defined as the relative difference between the rates $\Gamma$ of the $\Lambda_b^0$ decay and the CP-conjugated $\overline{\Lambda}_b^0$ decay,

$$\mathcal{A}_{CP} \equiv \frac{\Gamma(\Lambda_b^0 \to pK^-\pi^+\pi^-) - \Gamma(\overline{\Lambda}_b^0 \to \bar{p}K^+\pi^-\pi^+)}{\Gamma(\Lambda_b^0 \to pK^-\pi^+\pi^-) + \Gamma(\overline{\Lambda}_b^0 \to \bar{p}K^+\pi^-\pi^+)}. \tag{1}$$

Our experimental procedures are detailed in Methods.

According to the Standard Model, this asymmetry arises from the interference between the 'tree' and 'loop' quark-level amplitudes[23] of the $\Lambda_b^0$ baryon decay, which is mediated by the weak interaction, as illustrated by the Feynman diagrams in Fig. 1. These two complex amplitudes[24] are associated with phases (referred to as weak phases) derived from the products of CKM matrix elements $V_{ub}V_{us}^*$ and $V_{tb}V_{ts}^*$. The difference in the weak phases between the two amplitudes plays a crucial role in CP violation. Additionally, strong interactions between quarks can introduce a possible strong-phase difference between the two amplitudes. The weak phases change sign from $\Lambda_b^0$ to $\overline{\Lambda}_b^0$ decays, whereas the strong phases are the same. For a sizeable CP violation to occur, the two amplitudes must have similar magnitudes and substantial differences in both weak and strong phases. However, although the weak phases are defined by the CKM mechanism, the strong phases and magnitudes of the amplitudes depend on the process and are challenging to calculate due to low-energy strong-interaction effects[25]. Studies of multibody $B$-meson decays indicate that interactions among

final-state particles in the decay can significantly enhance the strong phase[26–28]. The $\Lambda_b^0 \to pK^-\pi^+\pi^-$ decay can proceed through a rich spectrum of hadrons, such as excited nucleons decaying to the $p\pi^+\pi^-$ final state, which may create the necessary conditions for the manifestation of significant CP asymmetries[29]. Moreover, the size of the CP asymmetry may vary across the phase space[30], which is defined in terms of two-body and three-body masses of the final states, thus allowing enlarged effects to be observed by selecting regions with appropriate contributions from hadronic resonances.

The CP asymmetry in the $\Lambda_b^0$ decay, as defined in equation (1), was inferred through the yield asymmetry between the numbers ($N$) of observed $\Lambda_b^0 \to pK^-\pi^+\pi^-$ and $\overline{\Lambda}_b^0 \to \bar{p}K^+\pi^-\pi^+$ decays, defined as

$$\mathcal{A}_N \equiv \frac{N(\Lambda_b^0 \to pK^-\pi^+\pi^-) - N(\overline{\Lambda}_b^0 \to \bar{p}K^+\pi^-\pi^+)}{N(\Lambda_b^0 \to pK^-\pi^+\pi^-) + N(\overline{\Lambda}_b^0 \to \bar{p}K^+\pi^-\pi^+)}. \tag{2}$$

As depicted in Fig. 1, the $\Lambda_b^0$ and $\overline{\Lambda}_b^0$ baryons in this study were produced from 2011 to 2018 in high-energy proton–proton ($pp$) collisions provided by the Large Hadron Collider (LHC) at CERN. The total integrated luminosity of the data was about 9 fb$^{-1}$. Beauty baryons from $pp$ collisions then decay into final-state particles, which are detected by the LHCb detector. The LHCb experiment was designed to study CP violation in particles containing $b$ or $c$ quarks. Detailed descriptions of the LHCb detector and its performance can be found in refs. 1,31.

Events were selected to reduce the background, primarily arising from random combinations of final-state particles. More details on event selection can be found in Methods. Because of its relatively long lifetime, the $\Lambda_b^0$ baryon travels a measurable distance before decaying, resulting in a decay vertex displaced from the $pp$ collision point. The final-state particles of the signal decay have a relatively high transverse momentum (the component of the momentum transverse to the beam direction, reflecting the large $\Lambda_b^0$ mass). These characteristics were exploited to suppress the background due to random combinations of $p$, $K^-$, $\pi^+$ and $\pi^-$ particles through a machine-learning technique implemented with a boosted-decision-tree classifier[32,33]. A background involving misidentified particles, such as the $\Lambda_b^0 \to p\pi^-\pi^+\pi^-$ decay, where a $\pi^-$ candidate is reconstructed as a $K^-$, was mitigated using particle identification (PID) information.

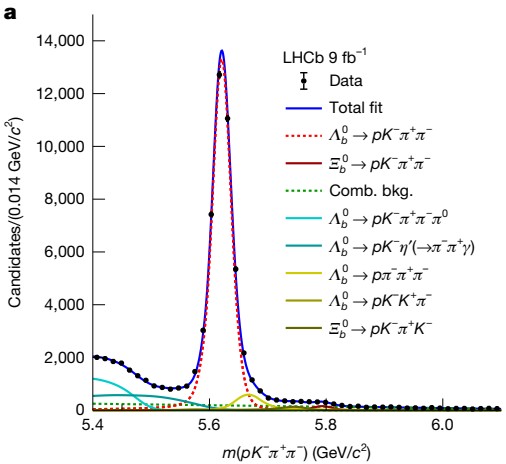

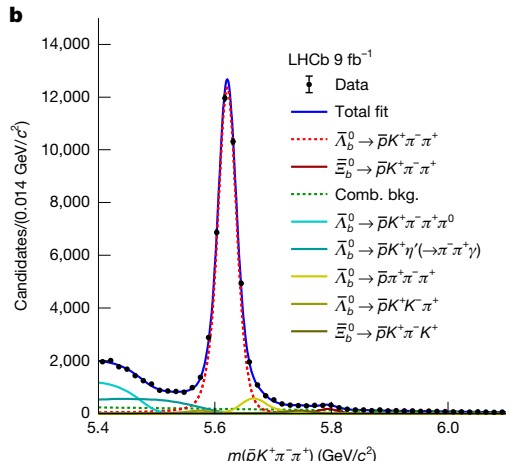

**Fig. 2 | Mass distributions together with the fitted projections. a,b,** Mass distributions for the signal channel: $\Lambda_b^0 \to pK^-\pi^+\pi^-$ (**a**) and $\overline{\Lambda}_b^0 \to \overline{p}K^+\pi^-\pi^+$ (**b**). The different components used in the fit are described in detail in Methods and listed in the legend. The area under a curve represents the yield of the corresponding component. Comb. bkg., combinatorial background.

The mass distributions of $\Lambda_b^0$ and $\overline{\Lambda}_b^0$ candidates, $m(pK^-\pi^+\pi^-)$ and $m(\overline{p}K^+\pi^-\pi^+)$, are displayed in Fig. 2. There are prominent peaks corresponding to the $\Lambda_b^0$ and $\overline{\Lambda}_b^0$ signal decays, along with remaining background components including the $\Xi_b^0 \to pK^-\pi^+\pi^-$ decay, random combinations of final-state particles, partially reconstructed $\Lambda_b^0$ decays and those involving misidentified particles. We performed extended unbinned maximum-likelihood fits to the mass spectra to extract the signal yields. In these fits, all identified contributions were modelled using empirical functions or distributions based on simulations, with the distribution for each component assumed to be identical for baryon and antibaryon decays. The yields were determined to be $N(\Lambda_b^0 \to pK^-\pi^+\pi^-) = (4.184 \pm 0.025) \times 10^4$ and $N(\overline{\Lambda}_b^0 \to \overline{p}K^+\pi^-\pi^+) = (3.885 \pm 0.023) \times 10^4$, giving a yield asymmetry of $\mathcal{A}_N = (3.71 \pm 0.39)\%$.

The measured yield asymmetry $\mathcal{A}_N$ differs from the CP asymmetry $\mathcal{A}_{CP}$ due to several biasing effects. First, due to the non-zero net baryon quantum number in $pp$ collisions, the production cross section of the $\Lambda_b^0$ baryon is slightly higher than that of the $\overline{\Lambda}_b^0$ baryon[34], resulting in a production asymmetry. Second, because particles and antiparticles behave differently when they interact with the detector material, which is made of matter rather than antimatter, a small detection asymmetry arises. These effects, collectively referred to as nuisance asymmetries, were measured to be around 1%, depending on the momenta of the beauty baryon or the final-state particles, and had to be subtracted from $\mathcal{A}_N$.

The decay $\Lambda_b^0 \to \Lambda_c^+\pi^-$ with $\Lambda_c^+ \to pK^-\pi^+$ was used as the control channel when subtracting the nuisance asymmetries. It proceeds through a single dominant quark-level process. Therefore, CP violation was not expected. Consequently, the yield asymmetry in the control channel was primarily due to the nuisance asymmetries, measured as $\mathcal{A}_N = (1.25 \pm 0.23)\%$. Mass distributions for the control channel are shown in Extended Data Fig. 1. The difference between nuisance asymmetries in the signal channel and the control channel was measured to be 0.01%, demonstrating the effective cancellation between the two decays. Details of the measurement of nuisance asymmetries are given in Methods.

The CP asymmetry of the signal decay was obtained from its yield asymmetry by subtracting the control-channel yield asymmetry and the difference in nuisance asymmetries, leading to the measurement:

$$\mathcal{A}_{CP} = (2.45 \pm 0.46 \pm 0.10)\%.$$

The first uncertainty arises from the sample sizes of both the signal and control channels, whereas the second is due to nuisance asymmetries and the choice of mass-fitting models for $\Lambda_b^0$ and $\overline{\Lambda}_b^0$. This CP asymmetry differs from zero by 5.2 standard deviations, marking the observation of CP violation. The robustness of the measurement was confirmed across different data collection periods, LHCb magnetic-field configurations, which affect the trajectory of charged particles, various event-selection scenarios, different momentum intervals for beauty baryons, among other factors. The results are consistent across the different subsamples and align with previous measurements that used a fraction of the data and different event selections[35].

The $\Lambda_b^0 \to pK^-\pi^+\pi^-$ decay occurred primarily through hadronic resonances that decayed into two or three final-state particles. Identified hadronic resonances include excited baryons in the $pK^-$, $p\pi^+$, $p\pi^-$ or $p\pi^+\pi^-$ mass spectra, denoted as $R(pK^-)$, $R(p\pi^+)$, $R(p\pi^-)$ and $R(p\pi^+\pi^-)$, respectively. Additionally, excited strange mesons, $R(K^-\pi^+\pi^-)$ and $R(K^-\pi^+)$, and light unflavoured mesons, $R(\pi^+\pi^-)$, were also observed. The production mechanisms for these resonances are complicated, and the associated strong phases and relative strengths of the tree and loop amplitudes are expected to vary among resonances. This variability led to differences in the CP asymmetry across the final-state phase space of the beauty-baryon decay. The global CP asymmetry reported above represents a measurement averaged over the entire phase space. To investigate the resonance contributions to the global CP violation, our analysis was performed across regions of the $\Lambda_b^0$ decay phase space, chosen based on their resonance compositions. Among the different possible resonance topologies, four made notable contributions to the $\Lambda_b^0$ decay and were selected for further measurements. Data corresponding to these decays were chosen according to relevant two-body or three-body masses. The local CP asymmetries between

**Table 1 | Measurements of CP asymmetries in four phase-space regions**

| Decay topology | Mass region (GeV/$c^2$) | $\mathcal{A}_{CP}$ |
|---|---|---|
| $\Lambda_b^0 \to R(pK^-)R(\pi^+\pi^-)$ | $m_{pK^-} < 2.2$ | $(5.3 \pm 1.3 \pm 0.2)\%$ |
| | $m_{\pi^+\pi^-} < 1.1$ | |
| $\Lambda_b^0 \to R(p\pi^-)R(K^-\pi^+)$ | $m_{p\pi^-} < 1.7$ | $(2.7 \pm 0.8 \pm 0.1)\%$ |
| | $0.8 < m_{\pi^+K^-} < 1.0$ | |
| | or $1.1 < m_{\pi^+K^-} < 1.6$ | |
| $\Lambda_b^0 \to R(p\pi^+\pi^-)K^-$ | $m_{p\pi^+\pi^-} < 2.7$ | $(5.4 \pm 0.9 \pm 0.1)\%$ |
| $\Lambda_b^0 \to R(K^-\pi^+\pi^-)p$ | $m_{K^-\pi^+\pi^-} < 2.0$ | $(2.0 \pm 1.2 \pm 0.3)\%$ |

The regions were selected using two-body or three-body masses.

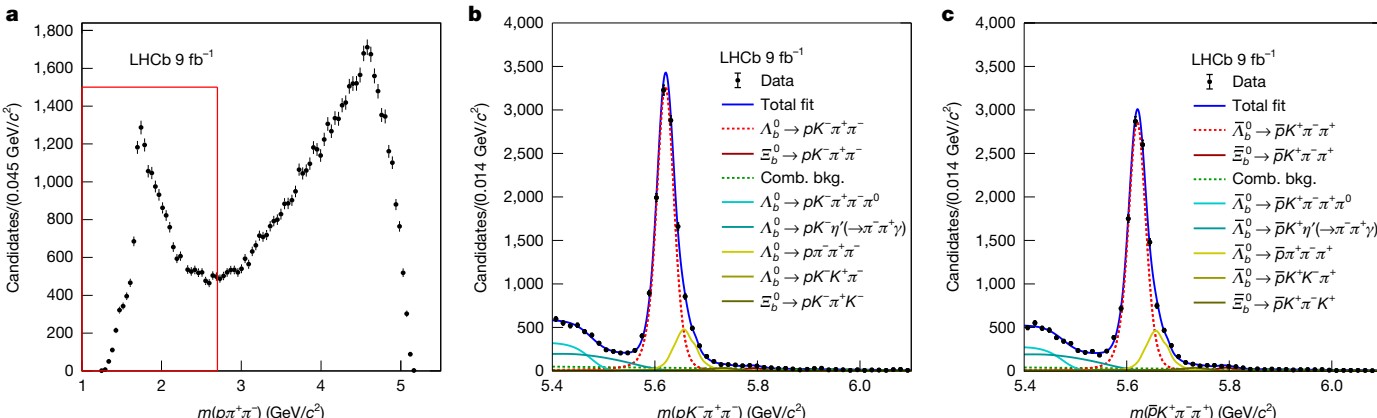

**Fig. 3 | Mass distributions in the $R(p\pi^+\pi^-)$ resonance phase space.**
**a**, Distribution of the $p\pi^+\pi^-$ mass including both $\Lambda_b^0$ and $\overline{\Lambda}_b^0$ candidates. The low-mass structure corresponds to excited nucleon resonances decaying to the $p\pi^+\pi^-$ final state, whereas the broad structure at higher masses arises from other decay processes of the $\Lambda_b^0$ baryon. **b,c**, Mass distributions of candidates within the region delimited by the red box in **a** are shown for $\Lambda_b^0 \rightarrow pK^-\pi^+\pi^-$ (**b**) and $\overline{\Lambda}_b^0 \rightarrow \overline{p}K^+\pi^-\pi^+$ (**c**) decays, together with the fitted projections and individual components.

$\Lambda_b^0$ and $\overline{\Lambda}_b^0$ decays in these regions were obtained like the global $\mathcal{A}_{CP}$ measurement.

A summary of the local phase-space decay topologies, selections and CP asymmetries is provided in Table 1. The CP asymmetry was most significant for the $\Lambda_b^0 \rightarrow R(p\pi^+\pi^-)K^-$ decay, with $\mathcal{A}_{CP} = (5.4 \pm 0.9 \pm 0.1)\%$, differing from zero by 6.0 standard deviations. The mass distributions for the $p\pi^+\pi^-$ system and the corresponding $\Lambda_b^0$ and $\overline{\Lambda}_b^0$ baryons are shown in Fig. 3 for $\Lambda_b^0 \rightarrow R(p\pi^+\pi^-)K^-$ decays. Mass distributions for other two-body or three-body systems, along with their corresponding $\Lambda_b^0$ and $\overline{\Lambda}_b^0$ baryons, are shown in Extended Data Figs. 2 and 3. The second most significant CP asymmetry was observed for the $\Lambda_b^0 \rightarrow R(pK^-)R(\pi^+\pi^-)$ decay, with $\mathcal{A}_{CP} = (5.3 \pm 1.3 \pm 0.2)\%$. The CP asymmetries for the other two decay topologies were not significant.

The CP asymmetry depends on decay topologies, so that the relative magnitudes or strong phases of the tree and loop amplitudes vary across the phase space. In general, the complicated hadronic effects pose important challenges for predicting CP asymmetries within the Standard Model. Various approaches have been proposed, such as using a model-independent investigation of angular distributions[36] or using scattering data to extract the hadronic amplitude[29]. An estimate of the CP asymmetry in $\Lambda_b^0 \rightarrow R(p\pi^+\pi^-)K^-$ decays made by applying this method using $\pi$–nucleon scattering data[37] aligns with the measurement in this work[29].

Each decay topology receives several resonant or non-resonant contributions that often overlap and interfere with each other. The intrinsic CP asymmetry can vary in magnitude and can even change sign between different contributions. As a result, the CP asymmetries reported here represent values averaged over the phase space. An investigation of the amplitude structure of this decay is left for future studies.

In summary, this Article presents, to our knowledge, the first observation of CP violation in baryon decays based on extensive samples of $\Lambda_b^0 \rightarrow pK^-\pi^+\pi^-$ and $\overline{\Lambda}_b^0 \rightarrow \overline{p}K^+\pi^-\pi^+$ decays collected with the LHCb detector. The measured CP asymmetry, $\mathcal{A}_{CP} = (2.45 \pm 0.46 \pm 0.10)\%$, with a significance of 5.2 standard deviations, reveals a difference in behaviour between baryonic matter and antimatter. We investigated various phase-space regions to better understand the source of the observed CP violation. In particular, the CP asymmetry was most pronounced in the region dominated by the resonant decays $\Lambda_b^0 \rightarrow R(p\pi^+\pi^-)K^-$, where it was measured to be $\mathcal{A}_{CP} = (5.4 \pm 0.9 \pm 0.1)\%$, which differs from zero by 6.0 standard deviations. This discovery strongly indicates that specific intermediate resonances play a key role in generating CP violation in $\Lambda_b^0$ decays. Furthermore, the generally small CP asymmetries in beauty-baryon decays imply that the dynamics in baryon decays are more complicated than in meson decays.

For instance, the CP asymmetries for various angular-momentum amplitudes of the same resonance may cancel[38]. This discovery of baryon decay asymmetry paves the way for further theoretical and experimental investigations into the nature of CP violation in baryon decays, potentially offering new constraints on scenarios beyond the Standard Model.

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

## LHCb Collaboration

R. Aaij[1], A. S. W. Abdelmotteleb[2], C. Abellan Beteta[3], F. Abudinén[2], T. Ackernley[4], A. A. Adefisoye[5], B. Adeva[6], M. Adinolfi[7], P. Adlarson[8,9], C. Agapopoulou[10], C. A. Aidala[5,11], Z. Ajaltouni[12], S. Akar[12], K. Akiba[1], P. Albicocco[13], J. Albrecht[14,87], F. Alessio[15], M. Alexander[9], Z. Aliouche[16], P. Alvarez Cartelle[17], R. Amalric[18], S. Amato[19], J. L. Amey[10], Y. Amhis[10], L. An[20], L. Anderlini[21], M. Andersson[3], A. Andreianov[111], P. Andreola[3], M. Andreotti[22], D. Andreou[5], A. Anelli[15,23,88], D. Ao[24], F. Archilli[25,89], M. Argenton[22], S. Arguedas Cuendis[15,26], A. Artamonov[111], M. Artuso[5], E. Aslanides[27], R. Ataíde Da Silva[28], M. Atzeni[29], B. Audurier[30], D. Bacher[31], I. Bachiller Perea[32], S. Bachmann[33], M. Bachmayer[28], J. J. Back[2], P. Baladron Rodriguez[6], V. Balagura[34], A. Balboni[22], W. Baldini[22], L. Balzani[14], H. Bao[24], J. Baptista de Souza Leite[4], C. Barbero Pretel[6,30], M. Barbetti[21], I. R. Barbosa[35], R. J. Barlow[16], M. Barnyakov[36], S. Barsuk[10], W. Barter[37], J. Bartz[5], J. M. Basels[38], S. Bashir[39], B. Batsukh[40], P. B. Battista[10], A. Bay[28], A. Beck[29], M. Becker[14], F. Bedeschi[41], I. B. Bediaga[42], N. A. Behling[14], S. Belin[6], K. Belous[111], I. Belov[43], I. Belyaev[44], G. Benane[27], G. Bencivenni[13], E. Ben-Haim[18], A. Berezhnoy[111], R. Bernet[3], S. Bernet Andres[45], A. Bertolin[36], C. Betancourt[3], F. Betti[37], J. Bex[17], Ia. Bezshyiko[3], O. Bezshyyko[10,47], J. Bhom[48], M. S. Bieker[14], N. V. Biesuz[22], P. Billoir[18], A. Biolchini[1], M. Birch[49], F. C. R. Bishop[32], A. Bitadze[16], A. Bizzeti[21,50], T. Blake[2], F. Blanc[28], J. E. Blank[14], S. Blusk[5], V. Bocharnikov[111], J. A. Boelhauve[14], O. Boente Garcia[34], T. Boettcher[51], A. Bohare[37], A. Boldyrev[111], C. S. Bolognani[52], R. Bolzonella[22], R. B. Bonacci[3], N. Bondar[15], A. Bordelius[15], F. Borgato[15,46], S. Borghi[16], M. Borsato[23,88], J. T. Borsuk[48,54], E. Bottalico[4], S. A. Bouchiba[28], M. Bovill[31], T. J. V. Bowcock[4], A. Boyer[15], C. Bozzi[22], J. D. Brandenburg[51,55], A. Brea Rodriguez[28], N. Breer[14], J. Brodzicka[48], A. Brossa Gonzalo[6,110], J. Brown[4], D. Brundu[56], E. Buchanan[37], L. Buonincontri[46,90], M. Burgos Marcos[1,52], A. T. Burke[16], C. Burr[15], J. S. Butter[17], J. Buytaert[15], W. Byczynski[15], S. Cadeddu[56], H. Cai[57,58], R. Calabrese[22,91], S. Calderon Ramirez[26], L. Calefice[59], S. Cali[13], M. Calvi[23,88], M. Calvo Gomez[45], P. Camargo Magalhaes[42,92], J. I. Cambon Bouzas[6], P. Campana[13], D. H. Campora Perez[1,52], A. F. Campoverde Quezada[24], S. Capelli[23], L. Capriotti[22], R. Caravaca-Mora[26], A. Carbone[36,93], L. Carcedo Salgado[6], R. Cardinale[43,93], A. Cardini[56], P. Carniti[23,88], L. Carus[33], A. Casais Vidal[29], R. Caspary[33], G. Casse[4], M. Cattaneo[15], G. Cavallero[15,22], S. Cavallini[22,91], S. Celani[15], S. Cesare[60,94], A. J. Chadwick[4], I. Chahrour[5,11], H. Chang[58,95], M. Charles[18], Ph. Charpentier[15], E. Chatzianagnostou[1], M. Chefdeville[32], C. Chen[17], S. Chen[40], Z. Chen[24], A. Chernov[48], S. Chernyshenko[61], X. Chiotopoulos[1,52], V. Chobanova[45,62], S. Chobanova[45,62], V. Chobanova[45,62], M. Chrzaszcz[48], A. Chubykin[111], V. Chulikov[13,44], P. Ciambrone[13], X. Cid Vidal[6], G. Ciezarek[15], P. Cifra[15], P. E. L. Clarke[37], M. Clemencic[15], H. V. Cliff[17], J. Closier[15], J. L. Cobbledick[16], V. Coco[15], J. Cogan[27], E. Cogneras[12], L. Cojocariu[63], S. Collaviti[28], P. Collins[15], T. Colombo[15], M. Colonna[14],

A. Comerma-Montells[59], L. Congedo[64], A. Contu[56], N. Cooke[9], C. Coronel[65], I. Corredoira[30], A. Correia[18], G. Corti[15], J. Cottee Meldrum[7], B. Couturier[15], D. C. Craik[3], M. Cruz Torres[42], E. Curras Rivera[28], R. Currie[37], C. L. Da Silva[51], S. Dadabaev[111], L. Dai[66,67], X. Dai[58], E. Dall'Occo[15], J. Dalseno[45,62], C. D'Ambrosio[15], J. Daniel[12], A. Danilina[111], P. d'Argent[64], G. Darze[19], A. Davidson[2], J. E. Davies[16], O. De Aguiar Francisco[16], C. De Angelis[56,96], F. De Benedetti[15], J. de Boer[1], K. De Bruyn[1,68], S. De Capua[16], M. De Cian[33], U. De Freitas Carneiro Da Graca[42,87], E. De Lucia[13], J. M. De Miranda[42], L. De Paula[19], M. De Serio[64,97], P. De Simone[13], F. De Vellis[14], J. A. de Vries[1,52], F. Debernardis[64], D. Decamp[32], V. Dedu[27], S. Dekkers[53], L. Del Buono[18], B. Delaney[29], H.-P. Dembinski[14], J. Deng[67], V. Denysenko[3], O. Deschamps[12], F. Dettori[56,96], B. Dey[15,69], P. Di Nezza[13], I. Diachkov[111], S. Didenko[111], S. Ding[5], L. Dittmann[33], V. Dobishuk[61], A. D. Docheva[9], C. Dong[58,95], A. M. Donohoe[70], F. Dordei[56], A. C. dos Reis[42], A. D. Dowling[5], W. Duan[58,71], P. Duda[48,54], M. W. Dudek[48], L. Dufour[15], V. Duk[72], P. Durante[15], M. M. Duras[48,54], J. M. Durham[51], O. D. Durmus[15,69], A. Dziurda[48], A. Dzyuba[111], S. Easo[73], E. Eckstein[74], U. Egede[53], A. Egorychev[111], V. Egorychev[111], S. Eisenhardt[37], E. Ejopu[16], L. Eklund[8,9], M. Elashri[65], J. Ellbracht[14], S. Ely[49], A. Ene[63], E. Epple[65], S. Escher[38], J. Esen[33], T. Evans[16], F. Fabiano[56], L. N. Falcao[42], Y. Fan[24], B. Fang[24], L. Fantini[15,72,98], M. Faria[28], K. Farmer[37], D. Fazzini[23,88], L. Felkowski[48,54], M. Feng[24,40], M. Feo[42], A. Fernandez Casani[75], M. Fernandez Gomez[6], A. D. Fernez[76], F. Ferrari[36,93], F. Ferreira Rodrigues[19], M. Ferrillo[3], M. Ferro-Luzzi[15], S. Filippov[111], R. A. Fini[64], M. Fiorini[22,93], M. Firlej[39], K. L. Fischer[31], D. S. Fitzgerald[5,11], C. Fitzpatrick[16], T. Fiutowski[39], F. Fleuret[34], M. Fontana[15], F. Foreman[16], R. Forty[15], D. Foulds-Holt[17], V. Franco Lima[19], M. Franco Sevilla[76], M. Frank[15], E. Franzoso[22,91], G. Frau[16], C. Frei[15], D. A. Friday[16], J. Fu[24], Q. Führing[14,17], Y. Fujii[53], T. Fulghesu[27], E. Gabriel[1], G. Galati[64], M. D. Galati[1], A. Gallas Torreira[6], D. Galli[36,93], S. Gambetta[37], M. Gandelman[19], P. Gandini[60], B. Ganie[16], H. Gao[24], R. Gao[31], T. Q. Gao[17], Y. Gao[67], Y. Gao[67], Y. Gao[67], L. M. Garcia Martin[28], P. Garcia Moreno[59], J. Garca Pardiñas[15], P. Gardner[76], K. G. Garg[67], L. Garrido[59], C. Gaspar[15], A. Gavrikov[46], L. L. Gerken[14], E. Gersabeck[16], M. Gersabeck[77], T. Gershon[2], S. Ghizzo[43,92], Z. Ghorbanimoghaddam[7], L. Giambastiani[46,90], F. I. Giasemis[18,99], V. Gibson[17], H. K. Giemza[78], A. L. Gilman[31], M. Giovannetti[3], A. Gioventù[59], L. Girardey[16,73], C. Giugliano[22,91], M. A. Giza[48], F. C. Glaser[10,33], V. V. Gligorov[15,18], C. Göbel[19,35], E. Golobardes[45], D. Golubkov[111], A. Golutvin[15,49], S. Gomez Fernandez[59], W. Gomulka[39], F. Goncalves Abrantes[31], M. Goncerz[48], G. Gong[58,95], J. A. Gooding[14], I. V. Gorelov[111], C. Gotti[23], E. Govorkova[29], J. P. Grabowski[74], L. A. Granado Cardoso[15], E. Graugés[59], E. Graverini[28,100], L. Grazette[2], G. Graziani[21], A. T. Grecu[63], L. M. Greeven[1], N. A. Grieser[65], L. Grillo[9], S. Gromov[111], C. Gu[34], M. Guarise[22], L. Guerry[12], V. Guliaeva[111], P. A. Günther[33], A.-K. Guseinov[28], E. Gushchin[111], Y. Guz[15,20], T. Gys[15], K. Habermann[74], T. Hadavizadeh[53], C. Hadjivasiliou[76], G. Haefeli[28], C. Haen[15], G. Hallett[2], M. M. Halvorsen[15], P. M. Hamilton[76], J. Hammerich[15], Q. Han[46], X. Han[15,33], S. Hansmann-Menzemer[33], L. Hao[24], N. Harnew[31], T. H. Harris[53], M. Hartmann[10], S. Hashmi[39], J. He[24,101], F. Hemmer[15], C. Henderson[65], R. D. L. Henderson[2,53], A. M. Hennequin[15], K. Hennessy[4], L. Henry[28], J. Herd[49], P. Herrero Gascon[33], J. Heuel[38], A. Hicheur[19], G. Hijano Mendizabal[3], J. Horswill[16], R. Hou[67], Y. Hou[12], N. Howarth[4], J. Hu[58,71], W. Hu[20], X. Hu[58,95], W. Huang[24], W. Hulsbergen[1], R. J. Hunter[2], M. Hushchyn[111], D. Hutchcroft[4], M. Idzik[39], D. Ilin[111], P. Ilten[65], A. Inglessi[111], A. Iniukhin[111], A. Ishteev[111], K. Ivshin[111], R. Jacobsson[15], H. Jage[38], S. J. Jaimes Elles[15,75,79], S. Jakobsen[15], E. Jans[1], B. K. Jashal[75], A. Jawahery[76], V. Jevtic[14], E. Jiang[76], X. Jiang[24,40], Y. Jiang[24], Y. J. Jiang[20], M. John[31], A. John Rubesh Rajan[70], D. Johnson[80], C. R. Jones[17], T. P. Jones[2], S. Joshi[78], B. Jost[15], J. Juan Castella[17], N. Jurik[15], I. Juszczak[48], D. Kaminaris[28], S. Kandybei[81], M. Kane[37], Y. Kang[58,95], C. Kar[12], M. Karacson[15], D. Karpenkov[111], A. Kauniskangas[28], J. W. Kautz[65], M. K. Kazanecki[48], F. Keizer[15], M. Kenzie[17], T. Ketel[1], B. Khanji[5], A. Kharisova[111], S. Kholodenko[15,41], G. Khreich[10], T. Kirn[38], V. S. Kirsebom[23,88], O. Kitouni[51], O. Klaver[82], N. Kleijne[41,102], K. Klimaszewski[78], M. R. Kmiec[78], S. Koliiev[61], L. Kolk[14], A. Konoplyannikov[20], P. Kopciewicz[15], R. Koppenburg[1], A. Korchin[81], M. Korolev[111], I. Kostiuk[61], O. Kot[61], S. Kotriakhova[56], A. Kozachuk[111], P. Kravchenko[111], L. Kravchuk[111], M. Kreps[2], P. Krokovny[111], W. Krupa[5], W. Krzemien[78], O. Kshyvanskyi[61], S. Kubis[48,54], M. Kucharczyk[48], V. Kudryavtsev[111], E. Kulikova[111], A. Kupsc[8,9], B. K. Kutsenko[27], I. Kyryllin[81], D. Lacarrere[15], P. Laguarta Gonzalez[59], A. Lai[56], A. Lampis[56], D. Lancierini[49], C. Landesa Gomez[6], J. J. Lane[53], R. Lane[7], G. Lanfranchi[13], C. Langenbruch[33], J. Langer[14], O. Lantwin[111], T. Latham[2], F. Lazzari[15,41,100], C. Lazzeroni[80], R. Le Gac[27], H. Lee[4], R. Lefèvre[12], A. Leflat[111], S. Legotin[111], M. Lehuraux[2], E. Lemos Cid[15], O. Leroy[27], T. Lesiak[48], E. D. Lesser[5], B. Leverington[33], A. Li[58,95], C. Li[58,95], C. Li[17], H. Li[58,71], J. Li[67], K. Li[67], L. Li[16], M. Li[67], P. Li[24], P.-R. Li[40,83], Q. Li[24,40], S. Li[67], T. Li[66,67], T. Li[58,71], Y. Li[67], Y. Li[40], Z. Lian[58,95], X. Liang[5], S. Libralon[75], C. Lin[24], T. Lin[73], R. Lindner[15], V. Linton[49], H. Linton[49], V. Lisovskyi[28], R. Litvinov[15,56], D. Liu[67], F. L. Liu[53], G. Liu[58,71], K. Liu[40,83], S. Liu[24,40], W. Liu[67], Y. Liu[37], Y. Liu[40,83], Y. L. Liu[49], G. Loachamin Ordonez[19,35], A. Lobo Salvia[59], A. Loi[56], T. Long[17], J. H. Lopes[19], A. Lopez Huertas[59], S. López Soliño[6], Q. Lu[34], C. Lucarelli[21,103], D. Lucchesi[46,90], M. Lucio Martinez[1,52], V. Lukashenko[1,61], Y. Luo[20], A. Lupato[46,104], E. Luppi[22,91], K. Lynch[70], X.-R. Lyu[24], G. M. Ma[58,95], S. Maccolini[14], F. Machefert[10], F. Maciuc[63], B. Mack[5], I. Mackay[31], L. M. Mackey[5], L. R. Madhan Mohan[17], M. J. Madurai[80], A. Maevskiy[111], D. Magdalinski[1], D. Maisuzenko[111], J. J. Malczewski[48], S. Malde[31], L. Malentacca[15], A. Malinin[111], T. Maltsev[111], G. Manca[56,96], G. Mancinelli[27], C. Mancuso[60], R. Manera Escalero[59], F. M. Manganella[25], D. Manuzzi[36], D. Marangotto[60], J. F. Marchand[32], R. Marchevski[28], U. Marconi[36], E. Mariani[18], S. Mariani[15], C. Marin Benito[59], J. Marks[33], A. M. Marshall[7], G. Martelli[72,98], G. Martellotti[44], L. Martinazzoli[15], M. Martinelli[23,88], D. Martinez Gomez[1,68], D. Martinez Santos[45,62], F. Martinez Vidal[75], A. Martorell i Granollers[45], A. Massafferri[42], R. Matev[15], A. Mathad[15], V. Matiunin[111], C. Matteuzzi[5], K. R. Mattioli[34], A. Mauri[49], E. Maurice[34], J. Mauricio[59], P. Mayencourt[28], J. Mazorra de Cos[75], M. Mazurek[78], M. McCann[49], T. H. McGrath[16], N. T. McHugh[9], A. McNab[16], R. McNulty[70], B. Meadows[65], G. Meier[14], D. Melnychuk[78], F. M. Meng[58,95], M. Merk[1,52], A. Merli[49], L. Meyer Garcia[76], D. Miao[24,40], H. Miao[24], M. Mikhasenko[14,84], D. A. Milanes[79,105], A. Minotti[23,88], E. Minucci[13], T. Miralles[12], B. Mitreska[14], D. S. Mitzel[14], A. Modak[73], L. Moeser[14], R. A. Mohammed[31], R. D. Moise[38], E. F. Molina Cardenas[5,11], T. Mombächer[15], M. Monk[2,53], S. Monteil[12], A. Morcillo Gomez[6], G. Morello[13], M. J. Morello[41,102], M. P. Morgenthaler[33], J. Moron[39], W. Morren[14], A. B. Morris[15], A. G. Morris[27], R. Mountain[5], H. Mu[58,95], Z. M. Mu[20], E. Muhammad[2], F. Muheim[37], M. Mulder[1,68], K. Müller[3], F. Muñoz-Rojas[26], R. Murta[49], V. Mytrochenko[81], P. Naik[4], T. Nakada[28], R. Nandakumar[73], T. Nanut[15], I. Nasteva[19], M. Needham[37], E. Nekrasova[111], N. Neri[60], S. Neubert[74], N. Neufeld[15], P. Neustroev[111], J. Nicolini[14], D. Nicotra[1,52], E. M. Niel[15], N. Nikitin[111], Q. Niu[40,83], P. Nogarolli[19], P. Nogga[74], C. Normand[7], J. Novoa Fernandez[6], G. Nowak[65], C. Nunez[5,11], H. N. Nur[9], A. Oblakowska-Mucha[39], V. Obraztsov[111], T. Oeser[38],

S. Okamura[22,91], A. Okhotnikov[111], O. Okhrimenko[61], R. Oldeman[56,96], F. Oliva[37], M. Olocco[14], C. J. G. Onderwater[1,52], R. H. O'Neil[15], D. Osthues[14], J. M. Otalora Goicochea[19], P. Owen[3], A. Oyanguren[75], O. Ozcelik[37], F. Paciolla[41,106], A. Padee[78], K. O. Padeken[74], B. Pagare[2], T. Pajero[15], A. Palano[13], M. Palutan[13], X. Pan[58,95], G. Panshin[40], L. Paolucci[2], A. Papanestis[15,73], M. Pappagallo[64,97], L. L. Pappalardo[22], C. Pappenheimer[65], C. Parkes[16], D. Parmar[14,84], B. Passalacqua[22,91], G. Passaleva[21], D. Passaro[15,41], A. Pastore[64], M. Patel[49], J. Patoc[31], C. Patrignani[36,93], A. Paul[5], C. J. Pawley[1,52], A. Pellegrino[1], J. Peng[24,40], M. Pepe Altarelli[13], S. Perazzini[36], D. Pereima[111], H. Pereira Da Costa[51], A. Pereiro Castro[6], P. Perret[12], A. Perrevoort[1,68], A. Perro[15,27], M. J. Peters[65], K. Petridis[7], A. Petrolini[43,93], J. P. Pfaller[65], H. Pham[5], L. Pica[41], M. Piccini[72], L. Piccolo[56], B. Pietrzyk[32], G. Pietrzyk[10], R. N. Pilato[4], D. Pinci[18], F. Pisani[15], M. Pizzichemi[15,23,88], V. Placinta[63], M. Plo Casasus[6], T. Poeschl[15], F. Polci[18], M. Poli Lener[13], A. Poluektov[27], N. Polukhina[111], I. Polyakov[16], E. Polycarpo[19], S. Ponce[15], D. Popov[15,24], S. Poslavskii[111], K. Prasanth[37], L. Prouve[45,62], E. Provenzano[56,96], V. Pugatch[61], G. Punzi[41,100], S. Qasim[3], Q. Q. Qian[20], W. Qian[24], N. Qin[58,95], S. Qu[58,95], R. Quagliani[15], R. I. Rabadan Trejo[2], J. H. Rademacker[7], M. Rama[41], M. Ramírez García[5,11], V. Ramos De Oliveira[19,35], M. Ramos Pernas[2], M. S. Rangel[19], F. Ratnikov[111], G. Raven[82], M. Rebollo De Miguel[75], F. Redi[60,104], J. Reich[7], F. Reiss[77], Z. Ren[24], P. K. Resmi[11], R. Ribalda Galvez[59], R. Ribatti[28], G. R. Ricart[30,34], D. Riccardi[41,102], S. Ricciardi[73], K. Richardson[29], M. Richardson-Slipper[37], K. Rinnert[4], P. Robbe[10,15], G. Robertson[9], E. Rodrigues[4], A. Rodriguez Alvarez[59], E. Rodriguez Fernandez[6], J. A. Rodriguez Lopez[18,79], E. Rodriguez Rodriguez[6], J. Roensch[14], A. Rogachev[111], A. Rogovskiy[73], D. L. Rolf[14], P. Roloff[15], V. Romanovskiy[65], A. Romero Vidal[6], G. Romolini[28], F. Ronchetti[28], T. Rong[20], M. Rotondo[13], S. R. Roy[33], M. S. Rudolph[5], M. Ruiz Diaz[33], R. A. Ruiz Fernandez[6], J. Ruiz Vidal[1,52], J. Ryzka[39], J. J. Saavedra-Arias[26], J. J. Saborido Silva[6], R. Sadek[34], N. Sagidova[111], D. Sahoo[15,69], N. Sahoo[80], B. Saitta[56,96], M. Salomoni[15,23,88], I. Sanderswood[75], A. Sanmartin[15], C. Santamarina Rios[6], M. Santimaria[13], L. Santoro[42], E. Santovetti[25], A. Saputi[15,22], D. Saranin[111], A. Sarnatskiy[1,68], G. Sarpis[37], M. Sarpis[77,85], C. Satriano[44,107], A. Satta[25], M. Saur[40,83], D. Savrina[111], H. Sazak[38], F. Sborzacchi[13,15], A. Scarabotto[14], S. Schael[38], S. Scherl[4], M. Schiller[9], H. Schindler[15], M. Schmelling[86], B. Schmidt[15], S. Schmitt[38], H. Schmitz[74], O. Schneider[28], A. Schopper[49], N. Schulte[14], S. Schulte[28], M. H. Schune[10], G. Schwering[38], B. Sciascia[13], A. Sciuccati[15], I. Segal[14,84], S. Sellam[6], A. Semennikov[111], T. Senger[3], M. Senghi Soares[82], A. Sergi[43,93], N. Serra[3], L. Sestini[21], A. Seuthe[14], Y. Shang[20], D. M. Shangase[5,11], M. Shapkin[111], R. S. Sharma[111], I. Shchemerov[111], L. Shchutska[28], T. Shears[4], L. Shekhtman[111], Z. Shen[1], S. Sheng[24,40], V. Shevchenko[111], B. Shi[24], Q. Shi[24], Y. Shimizu[10], E. Shmanin[36], R. Shorkin[111], J. D. Shupperd[5], R. Silva Coutinho[5], G. Simi[46,90], S. Simone[64,97], M. Singha[15,69], N. Skidmore[2], T. Skwarnicki[5], M. W. Slater[80], E. Smith[29], K. Smith[51], M. Smith[49], A. Snoch[1], L. Soares Lavra[37], M. D. Sokoloff[65], F. J. P. Soler[9], A. Solomin[7], A. Solovev[111], I. Solovyev[111], N. S. Sommerfeld[74], R. Song[53], Y. Song[58,95], Y. Song[20], Y. S. Song[20], F. L. Souza De Almeida[1], B. Souza De Paula[19], E. Spadaro Norella[43,93], E. Spedicato[36], J. G. Speer[14], E. Spiridenkov[111], P. Spradlin[9], V. Sriskaran[15], F. Stagni[15], M. Stahl[14,84], S. Stahl[15], S. Stanislaus[31], M. Stefaniak[51,55], E. N. Stein[15], O. Steinkamp[3], O. Stenyakin[111], H. Stevens[14], D. Strekalina[111], Y. Su[24], F. Suljik[31], J. Sun[56], L. Sun[58,95], D. Sundfeld[42], W. Sutcliffe[3], K. Swientek[39], F. Swystun[17], A. Szabelski[78], T. Szumlak[39], Y. Tan[58,95], Y. Tang[57,58], M. D. Tat[33], A. Terentev[111], F. Terzuoli[15,41,106], F. Teubert[15], E. Thomas[15], D. J. D. Thompson[80], H. Tilquin[49], V. Tisserand[12], S. T'Jampens[32], M. Tobin[15,40], L. Tomassetti[22,91], G. Tonani[60], X. Tong[20], T. Tork[60], D. Torres Machado[42], L. Toscano[14], D. Y. Tou[58,95], C. Trippl[45], G. Tuci[33], N. Tuning[1], L. H. Uecker[38], A. Ukleja[39], D. J. Unverzagt[33], A. Upadhyay[15,69], B. Urbach[37], A. Usachov[82], A. Ustyuzhanin[111], U. Uwer[33], V. Vagnoni[36], V. Valcarce Cadenas[6], G. Valenti[36], N. Valls Canudas[15], J. van Eldik[15], H. Van Hecke[51], E. van Herwijnen[49], C. B. Van Hulse[6,108], R. Van Laak[28], M. van Veghel[1], R. Vazquez Gomez[59], P. Vazquez Regueiro[6], C. Vázquez Sierra[45,62], S. Vecchi[22], J. J. Velthuis[7], M. Veltri[21,109], A. Venkateswaran[28], M. Verdoglia[56], M. Vesterinen[2], D. Vico Benet[31], P. Vidrier Villalba[59], M. Vieites Diaz[6], X. Vilasis-Cardona[45], E. Vilella Figueras[4], A. Villa[36], P. Vincent[18], B. Vivacqua[19], F. C. Volle[80], D. vom Bruch[27], N. Voropaev[111], K. Vos[1,52], C. Vrahas[37], J. Wagner[14], J. Walsh[41], E. J. Walton[2,53], G. Wan[20], A. Wang[24], C. Wang[33], G. Wang[67], H. Wang[40,83], J. Wang[20], J. Wang[58,95], J. Wang[57,58], M. Wang[15], N. W. Wang[24], R. Wang[7], X. Wang[67], X. Wang[58,71], X. W. Wang[49], Y. Wang[20], Y. W. Wang[40,83], Z. Wang[10], Z. Wang[58,95], Z. Wang[60], J. A. Ward[2,53], M. Waterlaat[15], N. K. Watson[80], D. Websdale[49], Y. Wei[20], J. Wendel[45,62], B. D. C. Westhenry[7], C. White[17], M. Whitehead[7], E. Whiter[80], A. R. Wiederhold[16], D. Wiedner[14], G. Wilkinson[31], M. K. Wilkinson[65], M. Williams[29], M. J. Williams[15], M. R. J. Williams[37], R. Williams[17], Z. Williams[7], F. F. Wilson[73], M. Winn[30], W. Wislicki[78], M. Witek[48], L. Witola[14], G. Wormser[10], S. A. Wotton[17], H. Wu[5], J. Wu[67], X. Wu[57,58], Y. Wu[17,20], Z. Wu[24], K. Wyllie[15], S. Xian[58,71], Z. Xiang[40], Y. Xie[77], T. X. Xing[60], A. Xu[41], L. Xu[58,95], L. Xu[58,95], M. Xu[2], Z. Xu[15], Z. Xu[24], Z. Xu[40], K. Yang[49], S. Yang[24], X. Yang[20], Y. Yang[24,40], Z. Yang[20], V. Yeroshenko[10], H. Yeung[16], H. Yin[67], X. Yin[24], C. Y. Yu[20], J. Yu[66,67], X. Yuan[40], Y. Yuan[24,40], E. Zaffaroni[28], M. Zavertyaev[86], M. Zdybal[48], F. Zenesini[36], C. Zeng[24,40], M. Zeng[58,95], C. Zhang[20], D. Zhang[67], J. Zhang[24], L. Zhang[58,95], S. Zhang[66,67], S. Zhang[31], Y. Zhang[20], Y. Zhang[58,95], Z. Zhang[58,95], Y. Zhao[33], A. Zhelezov[33], S. Z. Zheng[20], X. Z. Zheng[58,95], Y. Zheng[24], T. Zhou[20], X. Zhou[67], Y. Zhou[24], Y. Zhovkovska[2], L. Z. Zhu[24], X. Zhu[58,95], X. Zhu[67], V. Zhukov[38], J. Zhuo[75], Q. Zou[24,40], D. Zuliani[46,90] & G. Zunica[28]

[1]Nikhef National Institute for Subatomic Physics, Amsterdam, The Netherlands. [2]Department of Physics, University of Warwick, Coventry, UK. [3]Physik-Institut, Universität Zürich, Zurich, Switzerland. [4]Oliver Lodge Laboratory, University of Liverpool, Liverpool, UK. [5]Syracuse University, Syracuse, NY, USA. [6]Instituto Galego de Fsica de Altas Enerxas (IGFAE), Universidade de Santiago de Compostela, Santiago de Compostela, Spain. [7]H.H. Wills Physics Laboratory, University of Bristol, Bristol, UK. [8]Department of Physics and Astronomy, Uppsala University, Uppsala, Sweden. [9]School of Physics and Astronomy, University of Glasgow, Glasgow, UK. [10]Université Paris-Saclay, CNRS/IN2P3, IJCLab, Orsay, France. [11]University of Michigan, Ann Arbor, MI, USA. [12]Université Clermont Auvergne, CNRS/IN2P3, LPC, Clermont-Ferrand, France. [13]INFN Laboratori Nazionali di Frascati, Frascati, Italy. [14]Fakultät Physik, Technische Universität Dortmund, Dortmund, Germany. [15]European Organization for Nuclear Research (CERN), Geneva, Switzerland. [16]Department of Physics and Astronomy, University of Manchester, Manchester, UK. [17]Cavendish Laboratory, University of Cambridge, Cambridge, UK. [18]LPNHE, Sorbonne Université, Paris Diderot Sorbonne Paris Cité, CNRS/IN2P3, Paris, France. [19]Universidade Federal do Rio de Janeiro (UFRJ), Rio de Janeiro, Brazil. [20]School of Physics State Key Laboratory of Nuclear Physics and Technology, Peking University, Beijing, China. [21]INFN Sezione di Firenze, Florence, Italy. [22]INFN Sezione di Ferrara, Ferrara, Italy. [23]INFN Sezione di Milano-Bicocca, Milan, Italy. [24]University of Chinese Academy of Sciences, Beijing, China. [25]INFN Sezione di Roma Tor Vergata, Rome, Italy. [26]Consejo Nacional de Rectores (CONARE), San Jose, Costa Rica. [27]Aix Marseille Univ, CNRS/IN2P3, CPPM, Marseille, France. [28]Institute of Physics, Ecole Polytechnique Fédérale de Lausanne (EPFL), Lausanne, Switzerland. [29]Massachusetts Institute of Technology, Cambridge, MA, USA. [30]Centre d'Etudes de Saclay (CEA) IRFU Sarclay, Université Paris-Saclay, Gif-sur-Yvette, France. [31]Department of Physics, University of Oxford, Oxford, UK. [32]Université Savoie Mont Blanc, CNRS, IN2P3-LAPP, Annecy, France. [33]Physikalisches Institut, Ruprecht-Karls-Universität Heidelberg, Heidelberg, Germany. [34]Laboratoire Leprince-Ringuet, CNRS/IN2P3, Ecole Polytechnique, Institut Polytechnique de Paris, Palaiseau, France. [35]Pontifcia Universidade Católica do Rio de Janeiro (PUC-Rio), Rio de Janeiro, Brazil. [36]INFN Sezione di Bologna, Bologna, Italy. [37]School of Physics and Astronomy, University of Edinburgh, Edinburgh, UK. [38]I. Physikalisches Institut, RWTH Aachen University, Aachen, Germany. [39]AGH - University of Krakow, Faculty of Physics and Applied Computer Science, Kraków, Poland. [40]Institute of High Energy Physics (IHEP), Beijing, China. [41]INFN Sezione di Pisa, Pisa, Italy. [42]Centro Brasileiro de Pesquisas Fsicas (CBPF), Rio de Janeiro, Brazil. [43]INFN Sezione di Genova, Genoa, Italy. [44]INFN Sezione di Roma La Sapienza, Rome, Italy. [45]DS4DS, La Salle, Universitat Ramon Llull, Barcelona, Spain. [46]INFN Sezione di Padova, Padua, Italy. [47]Taras Schevchenko University of Kyiv, Faculty of Physics, Kyiv, Ukraine. [48]Henryk Niewodniczanski Institute of Nuclear Physics Polish Academy of Sciences, Kraków, Poland. [49]Imperial College London, London, UK. [50]Università di Modena e Reggio Emilia, Modena, Italy. [51]Los Alamos National Laboratory (LANL), Los Alamos, NM, USA. [52]Universiteit Maastricht, Maastricht, The Netherlands. [53]School of Physics and Astronomy, Monash University, Melbourne, Victoria, Australia. [54]Tadeusz Kosciuszko Cracow University of Technology, Cracow, Poland. [55]Ohio State University, Columbus, OH, USA. [56]INFN Sezione di Cagliari, Monserrato, Italy. [57]School of Physics and Technology, Wuhan University, Wuhan, China. [58]Department of Engineering Physics, Tsinghua University, Beijing, China. [59]ICCUB, Universitat de Barcelona, Barcelona, Spain. [60]INFN Sezione di Milano, Milan, Italy. [61]Institute for Nuclear Research of the National Academy of Sciences (KINR), Kyiv, Ukraine. [62]Universidade da Coruña, A Coruña, Spain. [63]Horia Hulubei National Institute of Physics and Nuclear Engineering, Magurele, Romania. [64]INFN Sezione di Bari, Bari, Italy. [65]University of Cincinnati, Cincinnati, OH, USA. [66]School of Physics and Electronics, Hunan University, Changsha City, China. [67]Institute of Particle Physics, Central China Normal University, Wuhan, China. [68]Van Swinderen Institute, University of Groningen, Groningen, The Netherlands. [69]Eotvos Lorand University, Budapest, Hungary. [70]School of Physics, University College Dublin, Dublin, Ireland. [71]Guangdong Provincial Key Laboratory of Nuclear Science, Guangdong-Hong Kong Joint Laboratory of Quantum Matter, Institute of Quantum Matter, South China Normal University, Guangzhou, China. [72]INFN Sezione di Perugia, Perugia, Italy. [73]STFC Rutherford Appleton Laboratory, Didcot, UK. [74]Universität Bonn - Helmholtz-Institut für Strahlen und Kernphysik, Bonn, Germany. [75]Instituto de Física Corpuscular, Centro Mixto Universidad de Valencia - CSIC, Valencia, Spain. [76]University of Maryland, College Park, MD, USA. [77]Physikalisches Institut, Albert-Ludwigs-Universität Freiburg, Freiburg, Germany. [78]National Center for Nuclear Research (NCBJ), Warsaw, Poland. [79]Departamento de Física, Universidad Nacional de Colombia, Bogota, Colombia. [80]School of Physics and Astronomy, University of Birmingham, Birmingham, UK. [81]NSC Kharkiv Institute of Physics and Technology (NSC KIPT), Kharkiv, Ukraine. [82]Nikhef National Institute for Subatomic Physics and VU University Amsterdam, Amsterdam, The Netherlands. [83]Lanzhou University, Lanzhou, China. [84]Ruhr Universitaet Bochum, Fakultaet f. Physik und Astronomie, Bochum, Germany. [85]Vilnius University, Vilnius, Lithuania. [86]Max-Planck-Institut für Kernphysik (MPIK), Heidelberg, Germany. [87]Present address: Lamarr Institute for Machine Learning and Artificial Intelligence, Dortmund, Germany. [88]Present address: Università degli Studi di Milano-Bicocca, Milan, Italy. [89]Present address: Università di Roma Tor Vergata, Rome, Italy. [90]Present address: Università di Padova, Padua, Italy. [91]Present address: Università di Ferrara, Ferrara, Italy. [92]Present address: Facultad de Ciencias Físicas, Madrid, Spain. [93]Present address: Università di Bologna, Bologne, Italy. [94]Present address: Università degli Studi di Milano, Milan, Italy. [95]Present address: Center for High Energy Physics, Tsinghua University, Beijing, China. [96]Present address: Università di Cagliari, Cagliari, Italy. [97]Present address: Università di Bari, Bari, Italy. [98]Present address: Università di Perugia, Perugia, Italy. [99]Present address: LIP6, Sorbonne Université, Paris, France. [100]Present address: Università di Pisa, Pisa, Italy. [101]Present address: Hangzhou Institute for Advanced Study, UCAS, Hangzhou, China. [102]Present address: Scuola Normale Superiore, Pisa, Italy. [103]Present address: Università di Firenze, Florence, Italy. [104]Present address: Università di Bergamo, Bergamo, Italy. [105]Present address: Universidad de Ingeniería y Tecnología (UTEC), Lima, Peru. [106]Present address: Università di Siena, Siena, Italy. [107]Present address: Università della Basilicata, Potenza, Italy. [108]Present address: Universidad de Alcalá, Alcalá de Henares, Spain. [109]Present address: Università di Urbino, Urbino, Italy. [110]Deceased: A. Brossa Gonzalo. [111]Unaffiliated: A. Andreianov, A. Artamonov, K. Belous, A. Berezhnoy, V. Bocharnikov, A. Boldyrev, A. Chubykin, S. Dadabaev, A. Danilina, I. Diachkov, S. Didenko, A. Dzyuba, A. Egorychev, V. Egorychev, S. Filippov, D. Golubkov, I. V. Gorelov, S. Gromov, V. Guliaeva, E. Gushchin, M. Hushchyn, D. Ilin, A. Inglessi, A. Iniukhin, A. Ishteev, K. Ivshin, D. Karpenkov, A. Kharisova, M. Korolev, A. Kozachuk, P. Kravchenko, L. Kravchuk, P. Krokovny, V. Kudryavtsev, E. Kulikova, O. Lantwin, A. Leflat, S. Legotin, A. Maevskiy, D. Maisuzenko, A. Malinin, T. Maltsev, M. Matiunin, E. Nekrasova, P. Neustroev, N. Nikitin, V. Obraztsov, A. Okhotnikov, D. Pereima, N. Polukhina, S. Poslavskii, F. Ratnikov, A. Rogachev, N. Sagidova, D. Saranin, D. Savrina, A. Semennikov, M. Shapkin, I. Shchemerov, L. Shekhtman, V. Shevchenko, R. Shorkin, A. Solovev, I. Solovyev, E. Spiridenkov, O. Stenyakin, D. Strekalina, A. Terentev, A. Ustyuzhanin, N. Voropaev.

# Methods

## Derivation of the CP asymmetry

The CP asymmetry arises from interference between the tree and loop processes. The total amplitude of the $\Lambda_b^0$ decay is the sum of the tree and loop amplitudes:

$$A(\Lambda_b^0) = |A_T|e^{+i\phi_T}e^{i\delta_T} + |A_L|e^{+i\phi_L}e^{i\delta_L}, \tag{3}$$

where $\phi_T$ ($\delta_T$) and $\phi_L$ ($\delta_L$) are the weak (strong) phases of the tree and loop processes, respectively, with $|A_T|$ and $|A_L|$ being their magnitudes. Similarly, the total amplitude for the $\overline{\Lambda}_b^0$ decay is given by

$$A(\overline{\Lambda}_b^0) = |A_T|e^{-i\phi_T}e^{i\delta_T} + |A_L|e^{-i\phi_L}e^{i\delta_L}. \tag{4}$$

Substituting into equation (1), where the decay rate $\Gamma$ is proportional to the squared amplitude, the CP asymmetry is obtained as

$$\mathcal{A}_{CP} = \frac{|A(\Lambda_b^0)|^2 - |A(\overline{\Lambda}_b^0)|^2}{|A(\Lambda_b^0)|^2 + |A(\overline{\Lambda}_b^0)|^2} = \frac{2\sin\Delta\delta\sin\Delta\phi}{|A_T/A_L| + |A_L/A_T| + 2\cos\Delta\delta\cos\Delta\phi}. \tag{5}$$

A sizeable $\mathcal{A}_{CP}$ requires $A_T$ and $A_L$ to have comparable magnitudes, along with notable differences in both the weak ($\Delta\phi$) and strong ($\Delta\delta$) phases.

## LHCb detector

The LHC at the CERN near Geneva is the world's largest particle accelerator. Constructed within a 27-km underground circular tunnel, the LHC accelerates two counterrotating proton beams to speeds near that of light, colliding them at designated interaction points to produce high-energy particles. The LHCb detector[1,31] is located at one of these interaction points to capture and analyse particles produced in the $pp$ collisions. Optimized to record decays of hadrons containing $b$ quarks, the LHCb detector is a forward spectrometer enabling a broad physics programme, including high-precision measurements of CP violation and searches for rare decay processes. Its tracking system reconstructs the trajectories of charged particles and measures their momenta with a relative uncertainty that varies from 0.5% at low momentum to 1.0% at 200 GeV/$c$. The tracking system also measures the particle impact parameter relative to the $pp$ interaction point with a resolution of $(15 + 29/p_T)$ μm, where $p_T$ is the transverse momentum of the particle in GeV/$c$. These capabilities enable precise vertex reconstruction and kinematic analyses, which are essential for distinguishing signal events from the background. Alongside the tracking system, the LHCb detector includes two ring-imaging Cherenkov detectors, a calorimeter system and a muon-detection system to provide PID information for final-state particles. Collectively, these components enable the LHCb experiment to rigorously test the Standard Model and search for new physics through precise measurements.

## Data and simulation samples

Measurements were performed using collision data collected by the LHCb experiment in $pp$ collisions at centre-of-mass energies of 7 TeV (2011) and 8 TeV (2012), referred to hereafter as the run 1 period, and 13 TeV (2015–2018), referred to as the run 2 period. Simulated $\Lambda_b^0$ decays were used in selecting events and studying mass distributions. In the simulation, $pp$ collisions were generated using Pythia[39,40], with a specific LHCb configuration[41]. Decays of unstable particles were described by EvtGen[42], in which the final-state radiation was generated using Photos[43]. The interaction of the generated particles with the detector and its response were implemented using the Geant4 toolkit[44,45], as described in ref. 46.

## Event selection

The online event selection for $b$-hadron decays was performed by a trigger system designed to retain beauty and charm hadrons of interest while rejecting the light-hadron background[47]. The system consisted of two parts: (1) a hardware-based first-level trigger, which selected hadrons, photons and electrons with high-energy deposits in the calorimeter as well as muons with high $p_T$ and (2) a software-based high-level trigger, which reconstructed and selected decays of interest. The software trigger required a two-, three- or four-track secondary vertex with a significant displacement from any $pp$ collision point, known as the primary vertex. At least one charged particle must have had a large $p_T$ and be inconsistent with originating from a primary vertex. A multivariate algorithm[48,49] was used to identify secondary vertices consistent with the decay of a $b$ hadron.

In the analysis, the $\Lambda_b^0$ baryon was reconstructed by combining four tracks identified as a proton, a kaon and two pions. Further selection criteria were applied to suppress the background while retaining most of the signal decays. The same selection requirements were applied to the $\Lambda_b^0 \to pK^-\pi^+\pi^-$ and $\overline{\Lambda}_b^0 \to \overline{p}K^+\pi^-\pi^+$ decays. To reduce the background from tracks originating at a primary vertex, a large impact parameter with respect to any primary vertex was required for each final-state track. The four tracks had to form a common vertex with a significant displacement from any primary vertex. Furthermore, the $\Lambda_b^0$ momentum, calculated from the final-state particles, was required to point back to the associated primary vertex.

A few categories of background were further suppressed to obtain a high-purity $\Lambda_b^0$ sample. Fake $\Lambda_b^0$ candidates, formed by random combinations of tracks identified as $p$, $K^-$, $\pi^+$ or $\pi^-$, were suppressed using a boosted-decision-tree multivariate classifier[48]. The classifier had been trained with a simulated sample for the signal and collision data from the high-mass sideband for the background, and it used information related to the large $\Lambda_b^0$ mass and long lifetime as well as the decay topology. The large $\Lambda_b^0$ mass resulted in a relatively high $p_T$ for final-state particles compared to those originating directly from $pp$ collisions. The long $\Lambda_b^0$ lifetime caused a displacement of the decay vertex from the primary vertex, leading to final-state particles with relatively large impact parameters. A second type of background arose from decays proceeding through intermediate charmed resonances. The final states of these charmed decays were like signal decays in that they formed well-reconstructed displaced vertices, and it was difficult to completely remove them through PID, kinematic or topological selections. Nevertheless, charmed resonances manifested as distinct peaks in the mass spectra of their decay products. For example, the $\Lambda_b^0 \to pD^0\pi^-$ decay with $D^0 \to K^-\pi^+$ had the same final state as the signal decay, but with the $K^-\pi^+$ mass peaking around the known $D^0$ mass[30]. By analysing these spectra, charmed resonances such as $D^0$, $D^+$, $\Lambda_c^+$ and $J/\psi$ were identified. Candidates near the charm mass peaks were excluded, which effectively suppressed the charmed background. Background arising from the misidentification of final-state particle species, such as a kaon misidentified as a proton or pion, was mitigated using a set of neural-network-based PID variables[50]. In addition, some misidentified particles originated from intermediate charmed resonances. By reconstructing the mass spectra with the appropriate particle masses, prominent resonance peaks were identified and excluded, effectively reducing this background contribution.

The otherwise excluded $\Lambda_b^0 \to \Lambda_c^+(\to pK^-\pi^+)\pi^-$ decay, which had the same final state as the signal channel, was the control channel. The CP asymmetry in the $\Lambda_b^0 \to \Lambda_c^+\pi^-$ decay was expected to be negligible in the Standard Model, as it is dominated by the tree-level $b \to c\overline{u}d$ transition, making it suitable for calibrating and cancelling nuisance asymmetries. All $\Lambda_b^0 \to \Lambda_c^+(\to pK^-\pi^+)\pi^-$ candidates were selected using the same criteria as imposed for the signal candidates, with the $pK^-\pi^+$ mass confined to a region centred around the known $\Lambda_c^+$ mass[30].

## Asymmetry measurement

The asymmetry was measured separately for run 1 and run 2 samples, from signal yield extraction to the evaluation of nuisance asymmetries and systematic uncertainties. The ratio of yields between run 1 and run 2 data was measured to be consistent with the estimate based on relative luminosity, cross section and efficiency. The signal yields from the two samples were then combined. The results of the nuisance asymmetries and systematic uncertainties were statistically averaged. This averaging used weights inversely proportional to the squared statistical uncertainties, which were determined from the respective mass fits. The yield asymmetries of the signal decay were measured to be $\mathcal{A}_N = (5.12 \pm 0.96)\%$ for run 1 and $\mathcal{A}_N = (3.42 \pm 0.43)\%$ for run 2, whereas those for the control mode were $\mathcal{A}_N = (1.08 \pm 0.55)\%$ for run 1 and $\mathcal{A}_N = (1.32 \pm 0.25)\%$ for run 2.

The signal yields for the $\Lambda_b^0 \to pK^-\pi^+\pi^-$ and $\overline{\Lambda}_b^0 \to \bar{p}K^+\pi^-\pi^+$ decays were determined from a simultaneous extended unbinned maximum-likelihood fit to their mass spectra. The signal shape was modelled as a combination of a Gaussian and two crystal ball functions[51], all with the same peak position. The parameters of the signal function were determined from simulated events and were fixed in the fit to collision data, except for the Gaussian width, the average width of the crystal ball functions and the peak position, which accounted for imperfections in the simulation. The results for these floated parameters were comparable for run 1 and run 2 data. Various background sources were modelled separately in the fit. The background from partially reconstructed events, specifically the $\Lambda_b^0 \to pK^-\pi^+\pi^-\pi^0$ decay where the $\pi^0$ meson was not reconstructed, was described by an ARGUS function[52]. The background from the $\Lambda_b^0 \to pK^-\eta'(\to\pi^+\pi^-\gamma)$ decay where the photon was not reconstructed, was modelled with a distribution obtained using fast-simulated decays[53].

The decays $\Lambda_b^0 \to p\pi^-\pi^+\pi^-$, $\Lambda_b^0 \to pK^-K^+\pi^-$ and $\Xi_b^0 \to pK^-\pi^+K^-$ could be incorrectly reconstructed as a $\Lambda_b^0 \to pK^-\pi^+\pi^-$ decay, with one final-state particle misidentified. Their mass distributions were modelled using a simulation. The $\Xi_b^0 \to pK^-\pi^+\pi^-$ decay was described by the same model used for the $\Lambda_b^0$ signal, but with the peak position shifted by the difference of the known $\Xi_b^0$ and $\Lambda_b^0$ baryon masses and with the width scaled by their mass ratio[54]. The combinatorial background, due to random combinations of final-state $p$, $K^-$, $\pi^+$ and $\pi^-$ hadrons, was modelled as a linear function.

For the control channel, the $\Lambda_b^0$ mass distribution was modelled using the same function as for the signal channel, with parameters determined independently from simulated $\Lambda_b^0 \to \Lambda_c^+(\to pK^-\pi^+)\pi^-$ decays. The background consisted of a combinatorial component modelled by a linear function, a partially reconstructed background from the $\Lambda_b^0 \to \Lambda_c^+\pi^-\pi^0$ decay modelled as an ARGUS function and a misidentified background from the $\Lambda_b^0 \to \Lambda_c^+K^-$ decay, which was modelled based on a sample of simulated decays.

In the simultaneous fit, the same probability density function for each component was used for both baryon and antibaryon decays, with the yields floated freely and independently for the two decays. The yield asymmetries obtained from the mass fits were affected by nuisance effects arising from asymmetries in the $b$-baryon production cross section as well as from the detection, reconstruction and selection of final-state particles. Most of these effects cancelled in the difference between the yield asymmetries of the signal and the control channels. Nevertheless, the difference in the $\Lambda_b^0$ or final-state kinematics between the signal and control channels led to an incomplete cancellation of nuisance asymmetries. To address this, the nuisance asymmetry difference was subtracted from the yield asymmetry difference after all such experimental asymmetries had been accounted for.

The origins and corrections for experimental asymmetries are outlined below. The initial $pp$ collision is a two-baryon system, which produces slightly more $\Lambda_b^0$ baryons than $\overline{\Lambda}_b^0$ baryons, resulting in an asymmetry that depends on their rapidity $y$ and their $p_T$ (ref. 34). The background-subtracted distributions of $y$ and $p_T$ for both the signal and control channels were first obtained using the sPlot technique[55]. Subsequently, the control-channel distributions were weighted to match those of the signal channel. As a result, the difference in production asymmetry between the signal channel and the kinematics-weighted control channel vanished.

Positively and negatively charged particles exhibit different behaviour when interacting with matter, resulting in different detection efficiencies, whose magnitude depends on the momentum of the particle. The final-state particles in this analysis included protons, kaons and pions. The proton-detection asymmetry as a function of momentum was measured in ref. 56. The kaon-detection asymmetry was studied using kaons from the $D^+ \to K^-\pi^+\pi^+$ decay[57], and the pion-detection asymmetry was investigated through $D^{*+} \to \pi^+D^0(\to K^-\pi^+\pi^-\pi^+)$ and $D^0 \to K_S^0\pi^+\pi^-$ decays[58,59]. Subsequently, the detection asymmetry for each final-state particle as a function of kinematics was averaged over the kinematic distribution in both the signal and control channels. Finally, the overall detection asymmetry was obtained by summing the detection asymmetry contributions from each final-state particle. The difference in detection asymmetries between the signal and control channels is presented in Extended Data Table 1, with the uncertainties estimated using pseudo-experiments. The same approach was applied to estimate the uncertainties from the PID asymmetry and trigger asymmetry discussed below.

The PID requirements applied to the final-state particles can introduce asymmetries between positively and negatively charged particles. PID efficiencies and asymmetries for final-state particles of $\Lambda_b^0$ and $\overline{\Lambda}_b^0$ decays were evaluated in bins of momentum $p$ and pseudorapidity $\eta$ using calibration samples of collision data[60,61]. Then the PID asymmetry for each final-state particle was averaged over the distribution in the $p$–$\eta$ plane for the signal and control channels. The difference in PID asymmetries between the signal and control channels is presented in Extended Data Table 1.

The determination of the hardware-trigger efficiency asymmetry between oppositely charged hadrons used the same methodology as for the PID asymmetry. The trigger efficiencies were studied separately for two categories of events: trigger on signal, where the trigger decision was based on the final state of the $\Lambda_b^0$ decay, and trigger independent of signal, where the trigger decision depended on other particles rather than those from the signal decay. The trigger-on-signal efficiency and its asymmetry for a final-state particle, as a function of its energy deposited in the calorimeter, were determined using the $\Lambda_b^0 \to \Lambda_c^+(\to pK^-\pi^+)\pi^-$ sample for protons and a $D^0 \to K^-\pi^+$ sample for kaons and pions. Conversely, the trigger-independent-of-signal efficiency and its asymmetry, as a function of the $\Lambda_b^0$ transverse momentum, were estimated using a control sample of $\Lambda_b^0$ decays[62]. The total trigger efficiency asymmetry was calculated as a weighted average of the asymmetries of the two categories of events. The difference between the signal and control channels is presented in Extended Data Table 1.

The total nuisance asymmetry difference is the sum of the detection asymmetry, PID asymmetry and trigger asymmetry differences between the signal and control channels. The corresponding uncertainties were calculated as the quadrature sum of the individual contributions. The results are also shown in Extended Data Table 1. The combined nuisance asymmetry difference for runs 1 and 2 was determined to be $(0.01 \pm 0.07)\%$.

## Systematic uncertainties

In addition to the systematic uncertainties arising from the nuisance asymmetries, further uncertainty is associated with the $\Lambda_b^0$ mass fit. This uncertainty was estimated by using alternative fitting models, with the largest variation in the yield asymmetry among these models assigned as the systematic uncertainty. For the alternative signal

models, the fixed parameters of the signal shapes were modified up or down by one standard deviation, the $\Lambda_b^0$ signal model was replaced with the sum of a Gaussian and a Hypatia function[63], and the difference between the $\Xi_b^0$ and $\Lambda_b^0$ masses, which was fixed during a fit, was varied by one standard deviation[30]. The combinatorial background distribution was changed from a linear function to an exponential function. The mass distribution of the $\Lambda_b^0 \to p\pi^-\pi^+\pi^-$ decay, reconstructed as the $\Lambda_b^0 \to pK^-\pi^+\pi^-$ decay, was modelled in the baseline fit using a simulated sample generated with the mixture of a uniform distribution in phase space and intermediate resonances. As an alternative, the mass distribution was modelled using simulated decays with only a uniform distribution in the phase space. Finally, the partially reconstructed background from the $\Lambda_b^0 \to pK^-\eta'(\to\pi^+\pi^-\gamma)$ decay was modelled using a simulated sample with other kinematic requirements applied. To evaluate the impact of the imperfect modelling of the partially reconstructed background from the $\Lambda_b^0 \to pK^-\pi^+\pi^-\pi^0$ decay, the $\Lambda_b^0$ mass-fitting range was changed from $5.40 < m(pK^-\pi^+\pi^-) < 6.10$ GeV/$c^2$ to $5.45 < m(pK^-\pi^+\pi^-) < 6.10$ GeV/$c^2$, where this background was reduced by around a factor of two. The results are presented in Extended Data Table 2.

The total systematic uncertainty was taken as the quadratic sum of the systematic uncertainties arising from nuisance asymmetries and the mass fit, as presented in Extended Data Table 2. The combined systematic uncertainty for runs 1 and 2 was 0.10%.

### Localized CP violation in the phase space

Local CP asymmetries were studied in four regions of phase space, selected based on the $\Lambda_b^0 \to pK^-\pi^+\pi^-$ decay topology. In total, seven resonance topologies are possible, with several hadronic resonances potentially contributing to each topology. Three of these topologies, namely $\Lambda_b^0 \to R(p\pi^+)K^-\pi^-$, $\Lambda_b^0 \to R(pK^-\pi^-)\pi^+$ and $\Lambda_b^0 \to R(pK^-\pi^+)\pi^-$, were strongly suppressed due to either the absence of relevant hadronic resonances or the flavour symmetry of the Standard Model[64], and these were not selected for localized CP measurements. Two-body and three-body mass distributions and the selected regions are shown in Extended Data Figs. 2 and 3, respectively. The four phase-space regions correspond to:

1. Two-body decays where the $pK^-$ and $\pi^+\pi^-$ two-body systems result from separate intermediate resonance decays, $R(pK^-)$ and $R(\pi^+\pi^-)$. The $pK^-$ mass was required to be less than 2.2 GeV/$c^2$. This area was dominated by excited $\Lambda$ resonances. The $\pi^+\pi^-$ mass was required to be less than 1.1 GeV/$c^2$. This area contained light unflavoured $f_0(500)$, $\rho(770)$ and $f_0(980)$ resonances and a non-resonant $\pi^+\pi^-$ component.
2. Two-body decays where the $p\pi^-$ and $\pi^+K^-$ two-body systems resulted from separate intermediate resonance decays, $R(p\pi^-)$ and $R(K^-\pi^+)$. We required that $m(p\pi^-) < 1.7$ GeV/$c^2$, delimiting a region dominated by excited nucleon ($N$) resonances, and also that $0.8 < m(\pi^+K^-) < 1.0$ GeV/$c^2$ or $1.1 < m(\pi^+K^-) < 1.6$ GeV/$c^2$, which contained mostly $K^{*0}$ resonances.
3. Three-body decay of excited $N^+$ resonances into the $p\pi^+\pi^-$ final state, $R(p\pi^-\pi^+)$. The requirement on the three-body mass was $m(p\pi^+\pi^-) < 2.7$ GeV/$c^2$.
4. Three-body decay of excited $K^-$ resonances into the $K^-\pi^+\pi^-$ final state, $R(K^-\pi^+\pi^-)$. The mass region $m(K^-\pi^+\pi^-) < 2.0$ GeV/$c^2$ included $K_1(1270)^-$, $K_1(1400)^-$ and $K^*(1410)^-$ resonances.

For each phase-space region, the CP asymmetry was measured with the same method as for the global $\mathcal{A}_{CP}$ result.

### Interpretation

In the $\Lambda_b^0 \to R(p\pi^+\pi^-)K^-$ decay, the $u$ quark produced from the $b \to u\bar{u}s$ process combines with the $ud$ quarks within the $\Lambda_b^0$ baryon to form the $R$ resonances, while the remaining $\bar{u}s$ quarks form the $K^-$ meson. A similar process occurs in the $\Lambda_b^0 \to R(K^-\pi^+\pi^-)p$ decay, where the $uud$ quarks form the proton and the $\bar{u}s$ quarks contribute

to the formation of the $R$ resonances. The measured CP asymmetry was different for the two decays, indicating that the tree and loop diagrams had different relative magnitudes or that there were strong phases. The $\Lambda_b^0 \to R(p\pi^-)R(K^-\pi^+)$ decay involves a $b \to d\bar{d}s$ transition, where the $d$ quark forms the $R(p\pi^-)$ baryon together with $ud$ quarks from the initial $\Lambda_b^0$ baryon. The remaining $\bar{d}s$ quarks form the $R(K^-\pi^+)$ meson. In this case, the tree amplitude does not contribute, and thus, CP symmetry was expected to hold. Conversely, the $\Lambda_b^0 \to R(pK^-)R(\pi^+\pi^-)$ decay represents a hybrid process involving both the $\Lambda_b^0 \to R(pK^-)f_0(980)(\to\pi^+\pi^-)$ process as well as non-resonant $\pi^+\pi^-$ contributions. The former was predominantly driven by the $b \to s\bar{s}s$ loop diagram, where an $s$ quark and the $ud$ quarks from the initial $\Lambda_b^0$ baryon hadronize into the $R(pK^-)$ baryon, and the rest forms the $f_0(980)$ hadron. The latter may arise from both tree and loop diagrams of the $b \to u\bar{u}s$ decay, where the $u\bar{u}$ quarks form the non-resonant $\pi^+\pi^-$ system, allowing CP violation to emerge.

### Significance and look-elsewhere effect

For the global $\mathcal{A}_{CP}$, the baseline method for evaluating significance was the $z$ score, which is the $\mathcal{A}_{CP}$ absolute value divided by its total uncertainty. For the local $\mathcal{A}_{CP}$, the preliminary significance was obtained by dividing $\mathcal{A}_{CP}$ by its total uncertainty. Subsequently, the look-elsewhere effect[65] was accounted for to correct for the increased probability of observing a significant result due to several measurements. The look-elsewhere effect was determined through pseudo-experiments, which also considered correlations among $\mathcal{A}_{CP}$ measurements in different phase-space regions.

### Data availability

LHCb data used in this analysis will be released according to the LHCb external data access policy, which can be downloaded from https://opendata.cern.ch/record/410/files/LHCb-Data-Policy.pdf. The raw data used for Figs. 1–3 and Extended Data Figs. 1–3 can be downloaded from https://cds.cern.ch/record/2927827. No access codes are required.

### Code availability

Software and code that is associated with this publication and that is publicly available is referenced within the publication content. Specific analysis software or code used to produce the results shown in the publication is preserved within the LHCb Collaboration internally and can be provided on reasonable request, provided it does not contain information that can be associated with unpublished results.

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

**Acknowledgements** We express our gratitude to our colleagues in the CERN accelerator departments for the excellent performance of the LHC. We thank the technical and administrative staff at the LHCb institutes. We acknowledge support from CERN and from the national agencies: ARC (Australia); CAPES, CNPq, FAPERJ and FINEP (Brazil); MOST and NSFC (China); CNRS/IN2P3 (France); BMBF, DFG and MPG (Germany); INFN (Italy); NWO (Netherlands); MNiSW and NCN (Poland); MCID/IFA (Romania); MICIU and AEI (Spain); SNSF and SER (Switzerland); NASU (Ukraine); STFC (United Kingdom); and DOE NP and NSF (USA). We acknowledge the computing resources provided by ARDC (Australia); CBPF (Brazil); CERN, IHEP and LZU (China); IN2P3 (France); KIT and DESY (Germany); INFN (Italy); SURF (Netherlands); Polish WLCG (Poland); IFIN-HH (Romania); PIC (Spain); CSCS (Switzerland); and GridPP (United Kingdom). We are indebted to the communities behind the various open-source software packages on which we depend. Individual groups or members have received support from Key Research Program of Frontier Sciences of CAS, CAS PIFI, CAS CCEPP, Fundamental Research Funds for the Central Universities and Sci. & Tech. Program of Guangzhou (China); Minciencias (Colombia); EPLANET, Marie Skłodowska-Curie Actions, ERC and NextGenerationEU (European Union); A*MIDEX, ANR, IPhU and Labex P2IO, and Région Auvergne-Rhône-Alpes (France); Alexander-von-Humboldt Foundation (Germany); ICSC (Italy); Severo Ochoa and María de Maeztu Units of Excellence, GVA, XuntaGal, GENCAT, InTalent-Inditex and Prog. Atracción Talento CM (Spain); SRC (Sweden); and the Leverhulme Trust, the Royal Society and UKRI (United Kingdom). Unaffiliated authors are affiliated with an institute formerly covered by a cooperation agreement with CERN.

**Author contributions** The LHCb experiment is being built, operated and maintained by the LHCb Collaboration. All authors contributed to the design, construction, deployment and operation of the detector, the data taking, the development of the reconstruction and simulation software, data processing and data analysis. The final manuscript was reviewed and approved by all authors.

**Funding** Open access funding provided by CERN (European Organization for Nuclear Research).

**Competing interests** The authors declare no competing interests.

**Additional information**
**Correspondence and requests for materials** should be addressed to X. Yang.

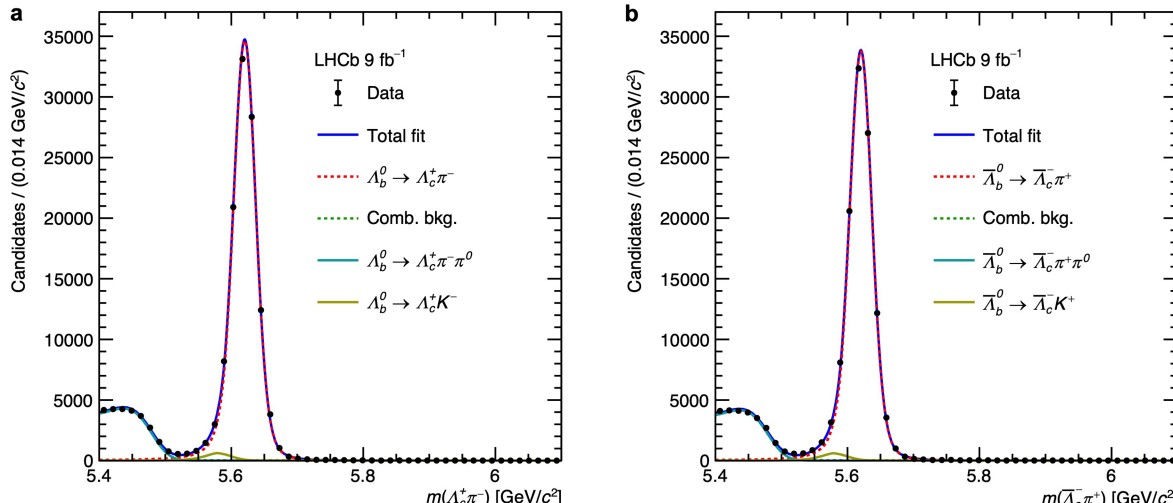

**Extended Data Fig. 1 | Mass distributions of the control channel together with the fit projections.** Displayed are the mass distributions for the control channel: (a) $\Lambda_b^0 \to \Lambda_c^+ \pi^-$, (b) $\overline{\Lambda}_b^0 \to \overline{\Lambda}_c^- \pi^+$.

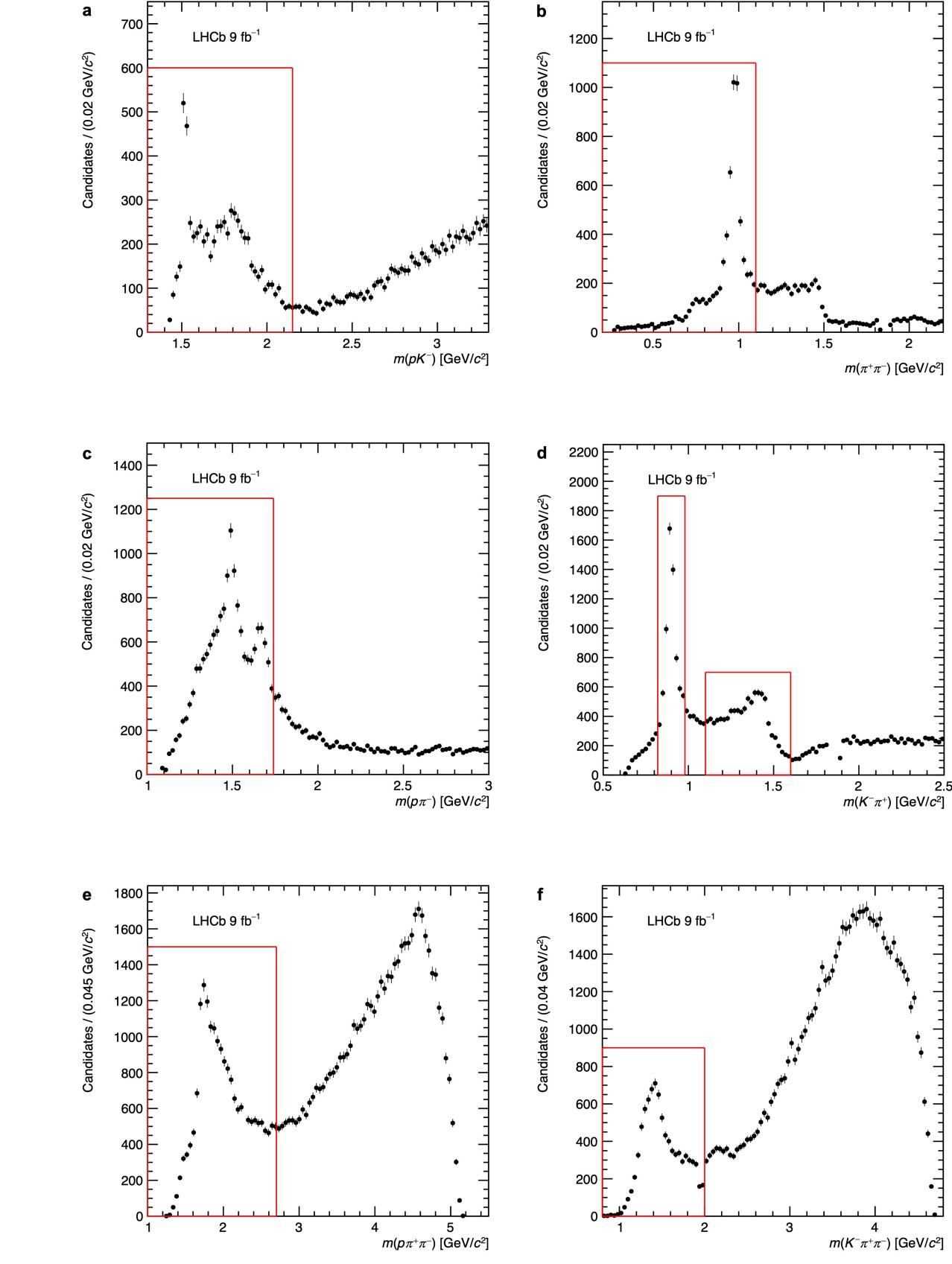

**Extended Data Fig. 2 | Distributions of two-body and three-body masses of final-state particles.** The mass distributions of (a) $pK^-$ and (b) $\pi^+\pi^-$, corresponding to the $\Lambda_b^0 \to R(pK^-)R(\pi^+\pi^-)$ phase-space region; (c) $p\pi^-$ and (d) $K^-\pi^+$, corresponding to the $\Lambda_b^0 \to R(p\pi^-)R(K^-\pi^+)$ phase-space region; (e) $p\pi^+\pi^-$, representing the $\Lambda_b^0 \to R(p\pi^+\pi^-)K^-$ phase-space region; and (f) $K^-\pi^+\pi^-$, representing the $\Lambda_b^0 \to R(K^-\pi^+\pi^-)p$ phase-space region. The $\Lambda_b^0$ and $\overline{\Lambda}_b^0$ samples are combined for the plots.

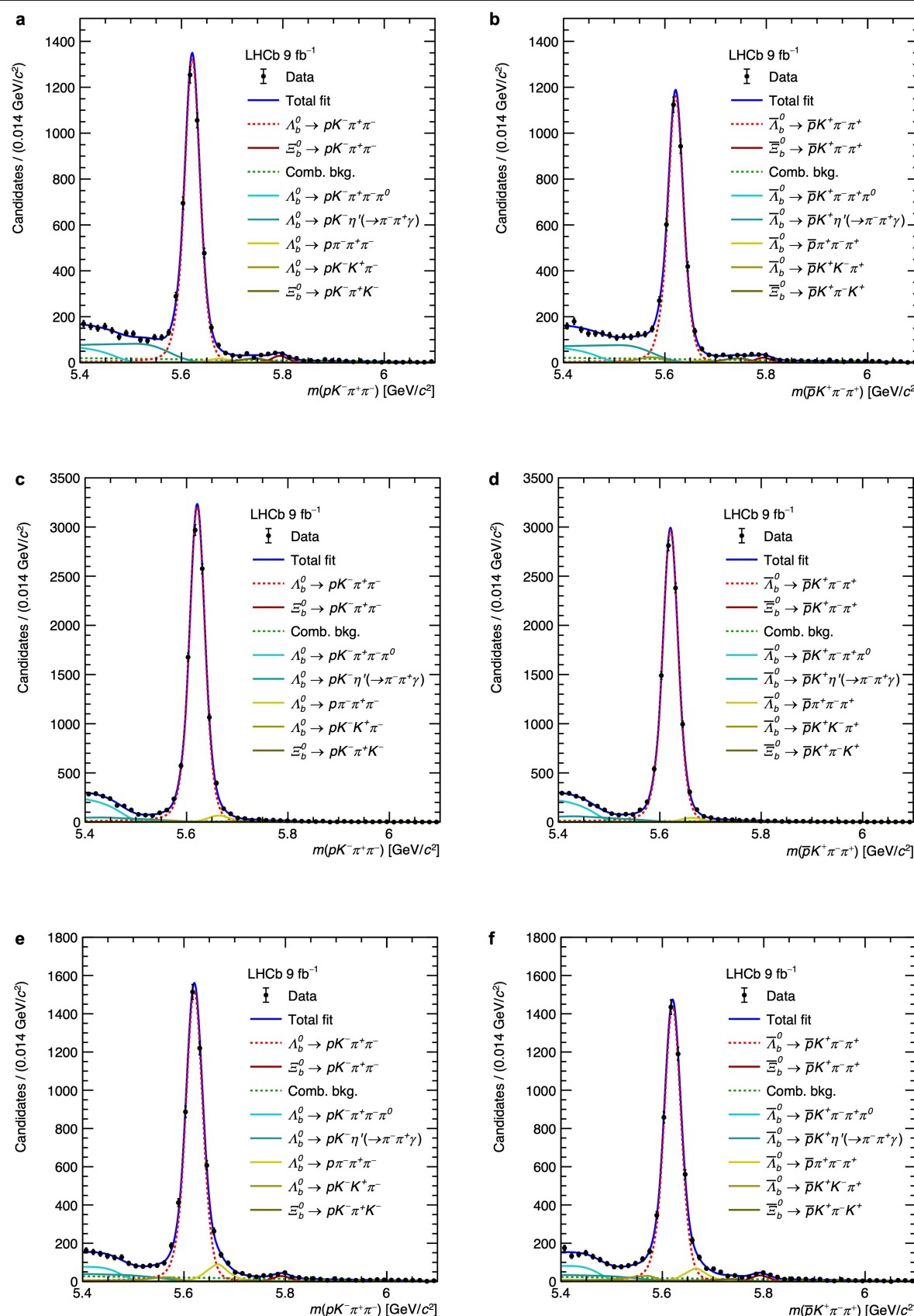

**Extended Data Fig. 3 | Mass distributions in regions of phase space with the fit projections also shown.** Mass distributions of $\Lambda_b^0 \to pK^-\pi^+\pi^-$ and $\overline{\Lambda}_b^0 \to \overline{p}K^+\pi^-\pi^+$ for (a, b) $\Lambda_b^0 \to R(pK^-)R(\pi^+\pi^-)$, (c, d) $\Lambda_b^0 \to R(p\pi^-)R(\pi^+K^-)$, and (e, f) $\Lambda_b^0 \to R(K^-\pi^+\pi^-)p$ decays.

**Extended Data Table 1 | Nuisance asymmetry differences between signal and control channels for Runs 1 and 2**

| Contribution | Run 1 | Run 2 |
|---|---|---|
| Detection asymmetry difference | (0.055 ± 0.128)% | (0.081 ± 0.050)% |
| PID asymmetry difference | (0.026 ± 0.141)% | (−0.028 ± 0.002)% |
| Trigger asymmetry difference | (−0.039 ± 0.029)% | (−0.050 ± 0.008)% |
| Total nuisance asymmetry difference | (0.042 ± 0.193)% | (0.003 ± 0.051)% |

**Extended Data Table 2 | Systematic uncertainties of the global $\mathcal{A}_{CP}$ measurement for Runs 1 and 2**

| Contribution | Run 1 | Run 2 |
|---|---|---|
| Nuisance asymmetry difference | 0.193% | 0.051% |
| Mass fit | 0.044% | 0.067% |
| Total systematic uncertainty | 0.198% | 0.084% |