## [Peer Review File · Nature]

Observation of charge-parity symmetry breaking in baryon decays

Corresponding Author: Ms Xueting Yang

Version 0:

Reviewer comments:

Referee #1

(Remarks to the Author)

This manuscript documents an important and topical result: the first observation of Charge-Parity (CP) symmetry violation in the decay of baryons. This is a very significant result, as it corresponds to the first observation of CP violation in baryons, since this phenomenon was first observed in mesons in the 1960's. The manuscript focuses in the multi-body decay of the Λ_b , a heavy baryon containing a b quark, which decays to a proton, a kaon and a pair of pions. Overall, this analysis resulted in a CP asymmetry of 2.45% with a 5.2 sigma significance, demonstrating the validity of the observation to the accepted standard for a discovery. CP violation in these decays arises from the interference between leading order and 1-loop matrix elements.

The Λ_b baryon decay can proceed through a number of resonant 2- and 3-particle states. This study went further to examine some of these decay modes, identifying two in which the CP asymmetry significance was particularly high, reaching 6 sigma. This not only gives further confidence in the overall asymmetry measurement, and evidence for CP-violation in barionic decays, but it also gives more detailed information about the relative importance of the tree and loop diagrams or of strong phases across the different phase space regions corresponding to each decay mode.

This result builds upon previous work by the LHCb experiment, for example focusing on Λ_b decays to a Λ and a pair of kaons, which reported evidence of CP-violation at a lower significance. But it is a clear piece of original work which represents significant advancement.

This result required the use of a substantial amount of data collected by the LHCb experiment at the LHC. It also required a complex and detailed analysis of that data, which is well documented in the present manuscript. The CP asymmetry is determined from the relative difference of the yields of Λ_b decays and their CP-conjugate mode, after correcting for biasing effects in production and detection with the help of a control channel where no CP violation is expected. The analysis is well summarised and justified in the manuscript.

The text is generally clear and complete, and the abstract, introduction and conclusions are lucid and appropriate. I have just a few suggestions and queries that I would like to ask the authors to consider:

- 1) The manuscript is generally very clear and complete, but it gives the impression of being written in 2 separate parts, with many useful experimental and analysis details given only in the second part. It might be useful to the reader to include, at some point e.g. around the beginning of page 2, a sentence saying that the detailed experimental methods are described later in the text (line 269). Another reference to the later Methods sections could be added to the end of the sentence starting in line 82;
- 2) Line 11: perhaps "dominance of matter" would be more clear and accurate than "persistence of matter";
- 3) The non-expert reader may be surprised that the Feynman diagrams in Figure 1 do not include the pion pair constituents in an obvious way. I would suggest trying a more complete description of the decay by augmenting the sentence "where the quarks eventually form..." in the figure caption;
- 4) Line 299 and later: for clarity I would advise the use of "collision data" or some equivalent term instead of "data" (simulation data are data, after all);
- 5) Line 312: I believe this is the first use of p_T ; please define it as transverse momentum;
- 6) Line 366 and following: I assume that the fits mentioned are performed separately for both run periods. Is this correct? Are

the fit results (distribution widths, for example) comparable, and are they consistent with simulation?

Finally, I would like to congratulate the authors on this work. It is my hope that this will be the first of a series of observations of CP violation in baryon decays.

Referee #2

(Remarks to the Author)

A/B A violation of the charge parity symmetry is one of the most fundamental properties of the laws of nature.

So far this effect has only been observed in the decays mesons with strange, charm and bottom quarks and was awarded for far two times with the nobel price.

In particular in B meson decays this asymmetry can be very pronounced, i.e. of the order of 50%. In contrast to that so far searches for CP violation in baryon decays yielded no non-vanishing result. The largest effect observed so far, is a deviation from zero by 3.1 standard deviations in the decay Λ_b to $\Lambda K^+ K^-$.

In this paper the LHCb collaboration presents the first ever measurement of a non-vanishing CP asymmetry in baryon decays with a statistical significance of 5.2 standard deviations, which is clearly an experimental breakthrough, that should be published to a broad audience.

A clear theory interpretation of the corresponding Λ_b to $p K^- \pi^+ \pi^-$ is beyond the scope of our current understanding of the hadronic dynamics, but I am sure that this discovery will trigger future experimental investigations of baryon decays with a simpler hadronic structure. Moreover the presented study of phase-space effects in this paper, might even guide upcoming theoretical developments.

C/D I cannot comment

E the conclusion are solid and robust

F for the current work: no suggested improvement

G The references are appropriate and give credit to the relevant previous work.

H The paper reads very well and makes the relevance of this work clear.

I strongly suggest the publication of the current draft.

A. Lenz

Referee #3

(Remarks to the Author)

This paper is based on LHCb run 1 and run 2 data and reports the first observation (5 sigma evidence) for CP violation in a b-baryon decay. In particular, the CP asymmetry in the decay $\Lambda_b \rightarrow p K^- \pi^+ \pi^-$ corresponding to the quark level transitions $b \rightarrow s u \bar{u}$ or $b \rightarrow s d \bar{d}$ is measured to be $(2.45 \pm 0.46(\text{stat}) \pm 0.10(\text{system}))\%$. Results for CP asymmetries in particular regions of the phase space are also provided.

This result is novel and significant as so far there has been only evidence at the 3 sigma level for CP violation in b-baryon decays and thus it deserves the visibility provided by a Nature publication. Let me add that the result itself is not unexpected nor does it require physics beyond the Standard Model to be explained. In fact, it is surprising that CP violation in b-baryon decays was not observed much earlier as the phenomenon was seen in B mesons already in 2001. It is likely that the complicated dynamics in b-baryon decays with different two-body and three-body resonances contributing with different amounts of tree and loop transitions and with different strong phase differences make it hard for a large net CP violation to arise and thus the enormous statistics of LHCb were necessary to see the effect. For the same reason I do not share the optimism of the authors that this result will help to constrain physics beyond the Standard Model. In fact, given the complexity of the hadronic physics involved, it will be hard to understand the result from first principles.

It is obvious that the authors have invested great care in making sure that the result is valid and robust. I have more detailed questions about specific points that I phrase below but in general the quality of the research is high and up to the standards of the field. Similar comments hold for the text of the manuscript: While I have specific comments for improvements, in general the article is well written, clear and properly references previous work.

In summary, I congratulate the authors and the LHCb collaboration in general on this nice result. I can recommend the article for publication after the authors have considered my detailed comments below.

Detailed comments:

Abstract and I. I. 45-47

I feel it is not clearly spelled out what is actually measured: It is said that the CP asymmetry in the decay Λ_b to $p K^- \pi^+ \pi^-$ is studied. However, you do not measure the direct decay as intermediate resonant contributions are included (and probably are even the dominant contribution to the CP asymmetry). On the other hand charmed resonances are vetoed (this is the control channel). What you actually measure becomes clear only after looking at the Feynman diagrams in Fig. 1.

So, I would write in the abstract and in the introduction that you study the CP asymmetry in Λ_b decays to $p K^- \pi^+ \pi^-$ corresponding to $b \rightarrow u$ or $b \rightarrow s$ quark level transitions.

Fig. 1

I understand that also $b \rightarrow s s \bar{s}$ can contribute when $f_0(980) \rightarrow \pi^+ \pi^-$ is formed. So, why is this not shown in the second Feynman diagram?

I. 102

No attempt is made to correct the yields for finite trigger and reconstruction efficiency. I'm ready to believe that most of the effect cancels in the ratio and is removed when subtracting the nuisance asymmetry. But shouldn't at least the difference in nuisance asymmetries between signal and control channel have some model-dependence based on which resonances actually contribute to the signal and to which extent? I would appreciate your comments on this point.

I. 125

The CP asymmetry is claimed to be 5.2 sigma which seems to be simply the central value divided by the quadratic sum of statistical and systematic errors. Has it been confirmed that errors are Gaussian to justify this approach? The systematic error is sub-dominant but if it would be, e.g., flat would this change anything to the 5 sigma claim?

I. 132 and following

I think it would be good to spell out at least a few of the hadronic resonances you have in mind. This would allow the Reader to better understand the mass region selections in Table 1.

I. 340

Please spell out which charm resonances you veto: Is it only $\Lambda_c \rightarrow p K^- \pi^+$ and $D^0 \rightarrow K^- \pi^+$ or are there also other modes (like $D^+ \rightarrow K^- \pi^+ \pi^-$).

I. 362

Can you please comment on the compatibility of the run 1 and 2 datasets? For the signal mode, the yield asymmetries differ by 1.5 sigma while for the control mode the difference is only 0.4 sigma. How statistically likely is this outcome?

I. 368

Is the mass fit using the same widths for Λ_b and Λ_b -bar? I understand that the widths are determined from data but are they constrained to be identical for particle and anti-particle? I would not be obvious to me that this must be the case.

I. 378

It is explained how the shapes of the different backgrounds are modeled but nothing is said about their normalization. Are the background yields freely floated in the mass fit or are they constrained somehow?

Extended Data Table 1

PID asymmetry difference error: It is 0.141% for run 1 but only 0.002% for run 2. How can you explain such a large difference?

Extended Data Table 2

It is odd that the mass fit systematics are larger for run 2 (0.067%) than for run 1 (0.044%). Otherwise uncertainties have always been smaller for run 2 (due to the larger integrated luminosity). Can you please comment.

We sincerely thank you for your insightful comments and suggestions, which have significantly contributed to improving the quality and readability of the manuscript.

Referee #1 (Remarks to the Author):

1. The manuscript is generally very clear and complete, but it gives the impression of being written in 2 separate parts, with many useful experimental and analysis details given only in the second part. It might be useful to the reader to include, at some point e.g. around the beginning of page 2, a sentence saying that the detailed experimental methods are described later in the text (line 269). Another reference to the later Methods sections could be added to the end of the sentence starting in line 82;

Reply: We agree with you. We have added reference to the Methods section at the beginning of page 2 (line 55), and line 87.

2. Line 11: perhaps “dominance of matter” would be more clear and accurate than “persistence of matter”;

Reply: We have replaced “persistence of matter” by “dominance of matter”.

3. The non-expert reader may be surprised that the Feynman diagrams in Figure 1 do not include the pion pair constituents in an obvious way. I would suggest trying a more complete description of the decay by augmenting the sentence “where the quarks eventually form...” in the figure caption;

Reply: We have updated the figure caption to provide a more complete description of the final-state particle formation. It now reads: *“The quarks in these processes eventually form $\$proton\$, \$Km\$, \$pip\$, and \$pim\$ particles, combined with additional $\$u\bar{u}\$ and $\$d\bar{d}\$ quark pairs created from the vacuum. The final states may also arise through intermediate hadronic resonances. The resulting hadrons are directly detected by the LHCb detector.”$$$*

4. Line 299 and later: for clarity I would advise the use of “collision data” or some equivalent term instead of “data” (simulation data are data, after all);

Reply: We have revised the manuscript to refer to “collision data” where appropriate.

5. Line 312: I believe this is the first use of p_T ; please define it as transverse momentum;

Reply: We note that p_T is firstly introduced and defined as transverse momentum earlier in line 292, so no changes were needed.

6. Line 366 and following: I assume that the fits mentioned are performed separately for both run periods. Is this correct? Are the fit results (distribution widths, for example) comparable, and are they consistent with simulation?

Reply: Indeed, the fits are performed separately for Run 1 and Run 2. This point is implicitly conveyed in L364. The fit results, including the widths of the signal distribution (i.e. experimental resolution), are comparable between the two run periods within uncertainties. Their relative yields are also consistent with expectations after accounting for luminosity, cross-section and selection efficiencies for Run 1 and Run 2. We have added "The fit is performed separately for Run~1 and Run~2 data, and the ratio of yields between Run~1 and Run~2 data is measured to be consistent with the estimation based on relative luminosity, cross-section, and efficiency." and "The results for these floated parameters are comparable for Run~1 and Run~2 data." in the relevant paragraph of the manuscripts.

Regarding the comparison with simulation: the fit to data uses the model developed from simulation, but the width parameters are left free. The fitted width differs from the MC width by about 5%, which is within expectations and consistent with other LHCb analyses. Allowing the width to float improves the agreement between the fits and data.

We sincerely thank you for your insightful comments and suggestions, which have significantly contributed to improving the quality and readability of the manuscript.

Referee #3 (Remarks to the Author):

Abstract and I. 45–47

I feel it is not clearly spelled out what is actually measured: It is said that the CP asymmetry in the decay Λ_b to $p K^- \pi^+ \pi^-$ is studied. However, you do not measure the direct decay as intermediate resonant contributions are included (and probably are even the dominant contribution to the CP asymmetry). On the other hand charmed resonances are vetoed (this is the control channel). What you actually measure becomes clear only after looking at the Feynman diagrams in Fig. 1. So, I would write in the abstract and in the introduction that you study the CP asymmetry in Λ_b decays to $p K^- \pi^+ \pi^-$ corresponding to $b \rightarrow u$ or $b \rightarrow s$ quark level transitions.

Reply: We agree with you. We have revised the abstract and the introduction (line47) to clarify that the studied CP asymmetry arises from Λ_b^0 decays to $p K^- \pi^+ \pi^-$ that proceed via $b \rightarrow u$ or $b \rightarrow s$ quark-level transitions, and that the measured final state includes contributions from intermediate resonances.

Fig. 1

I understand that also $b \rightarrow s s \bar{s}$ can contribute when $f_0(980) \rightarrow \pi^+ \pi^-$ is formed. So, why is this not shown in the second Feynman diagram?

Reply: Thank you for pointing this out. We have updated the second Feynman diagram to include the $s \bar{s}$ pair.

I. 102

No attempt is made to correct the yields for finite trigger and reconstruction efficiency. I'm ready to believe that most of the effect cancels in the ratio and is removed when subtracting the nuisance asymmetry. But shouldn't at least the difference in nuisance asymmetries between signal and control channel have some model-dependence based on which resonances actually contribute to the signal and to which extent? I would appreciate your comments on this point.

Reply: Indeed, most of the efficiency effect cancels out in the ratio between Λ_b and anti- Λ_b decays, and are further removed by additional subtraction of the nuisance asymmetry. A good cancellation between the signal and control channels is achieved due to the same set of final-state particles.

In practice, the nuisance asymmetry depends on the kinematics of the final-state particles, such as their momentum and rapidity. Since the kinematic distributions differ between the signal and control channels, due to different resonance contributions in the two decays, the nuisance asymmetries do not fully cancel. This is reflected in the residual asymmetries listed in Extended Data Table 1. Thus, the model dependence — arising from varying resonance contributions — is implicitly incorporated in our evaluation of the residual asymmetry and the corresponding uncertainty. Specifically, we use the kinematic distribution of each particle in the real data to calculate the nuisance asymmetry due to trigger and reconstruction efficiency, separately for the signal and control mode.

As said, no attempt is made to correct for the overall trigger and reconstruction efficiency, but only the difference between Λ_b and anti- Λ_b decays. This approach is widely used in LHCb analyses for multi-body decays for discovery of CP violation.

I. 125

The CP asymmetry is claimed to be 5.2 sigma which seems to be simply the central value divided by the quadratic sum of statistical and systematic errors. Has it been confirmed that errors are Gaussian to justify this approach? The systematic error is sub-dominant but if it would be, e.g., flat would this change anything to the 5 sigma claim?

Reply: Besides the method of simply dividing the central value by the total uncertainty, we also performed a profile likelihood scan for A_{CP} to derive the significance, and both methods give consistent results. The likelihood scan confirms that the statistical uncertainty behaves in a Gaussian-like way.

Regarding systematic uncertainties, the components associated with nuisance asymmetries are also approximately Gaussian, as they are largely driven by the statistics of the calibration samples. For the systematic uncertainty related to the mass fit (0.06% when averaging Run 1 and Run 2), we did not explicitly check its distribution, but it is assumed to be Gaussian. Even if this uncertainty were uniformly distributed (i.e., flat) rather than Gaussian, the impact would not lower the significance below 5σ .

To further ensure the robustness of the 5σ claim, we cross-checked using a more conservative approach. After performing the profile likelihood scan for A_{CP} , we tested the $A_{CP} = 0$ hypothesis by shifting A_{CP} by an amount corresponding to the total systematic uncertainty (0.10%) and computing the resulting change in two times the log-likelihood. This yielded a significance of 5.11σ — still above the 5σ threshold; it corresponds to a very conservative assumption about the distribution of the total systematic uncertainty. Thus, the significance is robust, especially given that the systematic uncertainty is small compared to the statistical uncertainty.

I. 132 and following

I think it would be good to spell out at least a few of the hadronic resonances you have in mind. This would allow the reader to better understand the mass region selections in Table 1.

Reply: We think that line 138 already lists all the hadronic resonance combinations we consider, and the Method section provides specific examples. Spelling them out here might result in some redundancy with the later content.

I. 340

Please spell out which charm resonances you veto: Is it only $\Lambda_c \rightarrow p K^- \pi^+$ and $D^0 \rightarrow K^- \pi^+$ or are there also other modes (like $D^+ \rightarrow K^- \pi^+ \pi^-$)?

Reply: We apply vetoes on charm resonances including $D^0 \rightarrow K^- \pi^+$, $\Lambda_c^+ \rightarrow p K^- \pi^+$, $\Lambda_c^+ \rightarrow p \pi^- \pi^+$, $D^+ \rightarrow K^- \pi^+ \pi^-$ and $J/\psi \rightarrow \pi^+ \pi^-$. The charm resonances have now been included in the manuscript.

I. 362

Can you please comment on the compatibility of the run 1 and 2 datasets? For the signal mode, the yield asymmetries differ by 1.5 sigma while for the control mode the difference is only 0.4 sigma. How statistically likely is this outcome?

Reply: The observed differences between run 1 and run 2 datasets are statistically compatible. I find that the two yield asymmetries (1.70 ± 1.05 (signal) and -0.24 ± 0.60 (control)) differ by 1.6 sigma and 0.4 sigma, corresponding to a p-value of 11%, and a p-value of about 69%. Assuming independence between the two tests, the joint probability is approximately 7.5%, indicating no significant inconsistency between Run 1 and Run 2.

I. 368

Is the mass fit using the same widths for Λ_b and Λ_b -bar? I understand that the widths are determined from data but are they constrained to be identical for particle and anti-particle? I would not be obvious to me that this must be the case.

Reply: Yes, the mass fit uses the same shape parameters, including widths, for Λ_b and anti- Λ_b .

We have checked that allowing the widths to float independently for Λ_b and anti- Λ_b decays yields consistent values within uncertainties. This approach is also well established in previous LHCb analyses. Moreover, the LHCb detector regularly reverses its magnetic field polarity, ensuring that the tracking momentum resolution is symmetric between positively and negatively charged particles averaging over both magnetic directions. This mitigates potential differences in mass resolution between Λ_b and anti- Λ_b .

I. 378

It is explained how the shapes of the different backgrounds are modeled but nothing is said about their normalization. Are the background yields freely floated in the mass fit or are they constrained somehow?

Reply: All the background yields are floated independently for Λ_b and anti- Λ_b decays in the mass fit

As a cross-check, we compare the ratios of the fitted yields of the misidentified backgrounds and the signal mode with the expectations based on known branching fractions from the

PDG, after correcting for the relative efficiencies. The two are consistent, which supports the reliability of the background modeling and normalization in the fit.

We replaced “allowed to differ between the two samples” with “with the yields allowed to float freely and independently for the two decays.” to more clearly describe the treatment of the yields.

Extended Data Table 1

PID asymmetry difference error: It is 0.141% for run 1 but only 0.002% for run 2. How can you explain such a large difference?

Reply: The Run 1 and Run 2 PID asymmetries are evaluated independently using calibration data. The larger uncertainty in Run 1 arises from the limited size of the calibration sample. In bins with low statistics, we adopt a conservative strategy: the asymmetry is taken from neighbouring bins, and the uncertainty is doubled to account for possible variations. This results in a relatively large uncertainty, ensuring that the true PID asymmetry is well covered. In contrast, the Run 2 calibration sample is much larger and has a much better coverage thanks to updated trigger schemes, allowing for a more precise determination with smaller associated uncertainties.

Extended Data Table 2

It is odd that the mass fit systematics are larger for run 2 (0.067%) than for run 1 (0.044%). Otherwise uncertainties have always been smaller for run 2 (due to the larger integrated luminosity). Can you please comment.

Reply: Most systematics (on nuisance asymmetries) are driven by detector-related uncertainties and benefit from larger calibration samples in Run 2. However, the mass fit systematic reflects imperfections of the fit model itself, such as sensitivity to various background shape choices. This uncertainty is assessed by varying the fit model, and its magnitude is not directly tied to sample size. There is no a priori reason to expect this uncertainty to be smaller for Run 2. In fact, the uncertainties for Run 1 and Run 2 are expected to be similar, which is consistent with the measured values.